# Dropping Just a Handful of Preferences Can Change Top Large Language Model Rankings

**Jenny Y. Huang**[1,2,*] **Yunyi Shen**[1,2,*] **Dennis Wei**[2,3] **Tamara Broderick**[1,2]

[1]Department of Electrical Engineering and Computer Science, Massachusetts Institute of Technology
[2]MIT-IBM Watson AI Lab        [3]IBM Research
{jhuang9,yshen99,tbroderick}@mit.edu, dwei@us.ibm.com

## Abstract

We propose a method for evaluating the robustness of widely used LLM ranking systems—variants of a Bradley–Terry model—to dropping a worst-case very small fraction of preference data. Our approach is computationally fast and easy to adopt. When we apply our method to matchups from popular LLM ranking platforms, including Chatbot Arena and derivatives, we find that the rankings of top-performing models can be remarkably sensitive to the removal of a small fraction of preferences; for instance, dropping just $0.003\%$ of human preferences can change the top-ranked model on Chatbot Arena. Our robustness check identifies the specific preferences most responsible for such ranking flips, allowing for inspection of these influential preferences. We observe that the rankings derived from MT-bench preferences are notably more robust than those from Chatbot Arena, likely due to MT-bench's use of expert annotators and carefully constructed prompts. Finally, we find that neither rankings based on crowdsourced human evaluations nor those based on LLM-as-a-judge preferences are systematically more sensitive than the other.

## 1 Introduction

Open evaluation platforms like Chatbot Arena (Chiang et al., 2024a) have, in large part due to their openness, become a gold standard for assessing the capabilities of leading LLMs via human preference. These open platforms are now widely used by top LLM developers and companies (The Wall Street Journal, 2024; Bloomberg, 2025) to evaluate and design new models and benchmarks (Grattafiori et al., 2024; Hui et al., 2024). Such platforms rely on crowdsourced pairwise battles and human votes to compute model rankings (Lee et al., 2023; Bai et al., 2022).

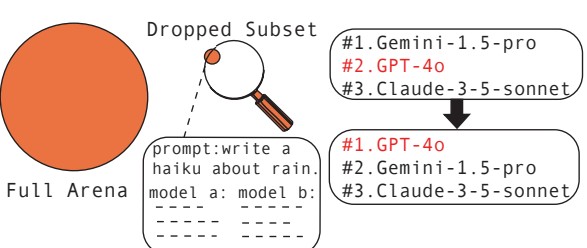

Figure 1: Our method (i) tests whether AI leaderboard rankings remain stable upon dropping small fractions of data and (ii) pinpoints the specific data points (e.g., preferences) that drive ranking flips.

At the heart of these preference-based evaluation pipelines is the Bradley–Terry (BT) model (Bradley & Terry, 1952), which is widely used to rank LLMs based on human feedback (Chiang et al., 2024a). The BT model is also used to train reward models for RLHF (Ouyang et al., 2022; Touvron et al., 2023; Xu et al., 2024; Sun et al., 2025) and route queries to the most appropriate LLM or inference-time scaling strategy (Damani et al., 2025).

A growing body of work has called into question the trustworthiness of LLM leaderboards, showing that they are vulnerable to adversarial attacks: a few hundred injected votes can change top rankings

---

[*]Equal Contribution

on Chatbot Arena (Min et al., 2025), attackers can identify model outputs to systematically upvote or downvote targets (Huang et al., 2025b), LLM-judges can be easily gamed (Zheng et al., 2025; Raina et al., 2024), and issues such as data leakage or selective reporting further undermine leaderboard reliability (Singh et al., 2025).

In this work, we study a different type of untrustworthiness of LLM ranking systems. That is: *"Will the top rankings from LLM-evaluation platforms change upon dropping a very small fraction of the human (or AI) preference evaluations?"* A positive answer would raise concerns about the stability and generalizability of rankings produced by such systems. Our notion of non-robustness differs from those of Min et al. (2025); Huang et al. (2025b); Zhao et al. (2025) in two major respects. First, it occurs at a different place in the process, at the data analysis step after data has been collected (including from malicious or apathetic users). Second, it does not require adversarial intent.

Our question posed above motivates the need for a systematic way to assess the robustness of top rankings in BT-based evaluation systems to worst-case data dropping. However, a brute-force combinatorial search over all possible small subsets of data would be computationally infeasible for large-scale platforms like Chatbot Arena. So we instead turn to a recent line of works from statistics and theoretical computer science that design algorithms for assessing whether data analyses are robust to dropping a small, worst-case fraction of data points (Broderick et al., 2020; Kuschnig et al., 2021; Moitra & Rohatgi, 2023; Freund & Hopkins, 2023; Shiffman et al., 2023; Nguyen et al., 2024; Huang et al., 2025a; Rubinstein & Hopkins, 2025). One such method, the Approximate Maximum Influence Perturbation (AMIP), estimates how much a statistic of interest could change if a worst-case subset of the data were dropped (Broderick et al., 2020). We extend these ideas to develop a fast approximation method for assessing the robustness of rankings from LLM evaluation systems to worst-case data-dropping.

We apply our method to assess several popular LLM ranking platforms, including Chatbot Arena and derivatives (Chiang et al., 2024a; Zheng et al., 2023; Miroyan et al., 2026; Vichare et al., 2025; Chou et al., 2025) and find most to be non-robust to dropping a very small fraction of votes.

In Section 2, we formalize the setup for assessing worst-case data-dropping robustness in BT-based ranking systems, and in Section 3 we introduce a computationally efficient method for assessing this form of robustness in practice (Figure 1). In Section 4, we apply our robustness assessment method to investigate the robustness of several LLM leaderboards.

## 2 SETUP

**Human preference data.** We consider a preference-based ranking system akin to Chatbot Arena (Chiang et al., 2024a). There are in total $M$ language models. Any user can submit a prompt to be answered by a pair of language models. Let the $n$th such prompt be sent to models $i_n$ and $j_n$ for $i_n, j_n \in [M] := \{1, \ldots, M\}$ with $i_n \neq j_n$. The user then determines if the response from model $i_n$ is better than that of model $j_n$, or is tied. Suppose there are in total $N$ such comparisons; the $n$th comparison can be seen as a tuple $(i_n, j_n, y_n)$, with $y_n \in \{W, L, T\}$ for whether in the $n$th match, model $i_n$ is preferred over model $j_n$ (a win, $W$), $j_n$ is preferred over $i_n$ (a loss, $L$), or the two models are similar (a tie, $T$). From a collection of preference data, the goal is to rank the language models.

**Ranking with the (unweighted) Bradley–Terry model.** The Bradley–Terry (BT) model is a classical statistical model used to rank players from *binary* match outcomes when there are only wins and losses, $y_n \in \{W, L\}$. In this model, each player (e.g., language model), $i$, is associated with a *BT score*, $\theta_i$, and the outcomes are modeled as

$$I_{y_n=W} \sim \text{Bernoulli}(\sigma(\theta_{i_n} - \theta_{j_n})), \tag{1}$$

where the sigmoid function $\sigma(x) = 1/(1 + e^{-x})$ and $I$ is the indicator function. Note, since the "winning" probability depends on the difference between two players' scores rather than on their raw scores, the scores are identified only up to a constant additive term. There are different ways to avoid this identifiability problem (Wu et al., 2022). Chatbot Arena chooses to set `mixtral-8x7b-instruct-v0.1` as the reference model, assigning it a fixed score of 1,114. Chatbot Arena computes the BT-scores (i.e., the estimates of $\boldsymbol{\theta} = (\theta_1, \ldots, \theta_M)$) for the unweighted

BT-model by maximum likelihood,

$$\hat{\boldsymbol{\theta}} := \arg\max_{\boldsymbol{\theta}:\theta_1=0} \sum_{n=1}^{N} \left( I_{y_n=W} \log \sigma(\theta_{i_n} - \theta_{j_n}) + I_{y_n=L} \log(1 - \sigma(\theta_{i_n} - \theta_{j_n})) \right). \tag{2}$$

Finally, the *rank* of a model is its position in the sorted list of models, $(\widehat{\theta}_{(1)}, \dots, \widehat{\theta}_{(M)})$, ordered by their scores in descending order, so that $\widehat{\theta}_{(1)}$ corresponds to the top-scoring and top-ranked model.[1]

**Ranking with the weighted Bradley–Terry model to handle ties.** The classic BT model cannot handle ties. To handle ties, Chatbot Arena adds weights to Equation (2), counting a tie as both a win and a loss (Chiang et al., 2024a).[2] In the weighted BT model, one specifies a weight for wins and losses, $w_{WL}$, and a weight for ties $w_T$. That is, we estimate BT scores by maximizing the weighted likelihood,

$$\hat{\boldsymbol{\theta}} := \arg\max_{\boldsymbol{\theta}:\theta_1=0} \sum_{n=1}^{N} \Big[ w_{WL} I_{y_n=W} \log \sigma(\theta_{i_n} - \theta_{j_n}) + w_{WL} I_{y_n=L} \log(1 - \sigma(\theta_{i_n} - \theta_{j_n}))$$
$$+ w_T I_{y_n=T} \left\{ \log \sigma(\theta_{i_n} - \theta_{j_n}) + \log(1 - \sigma(\theta_{i_n} - \theta_{j_n})) \right\} \Big]. \tag{3}$$

As done on Chatbot Arena, we use $w_{WL} = 2$ and $w_T = 1$. This choice can be interpreted as each win or loss counting as two matches of the same outcome, and a tie counting as one win and one loss. They also suggested an alternative treatment of dropping all ties and using the unweighted BT model, which corresponds to $w_{WL} = 1$ and $w_T = 0$.

**Postprocessing in Chatbot Arena.** Chatbot Arena applies a linear transformation to the learned BT scores (Chiang et al., 2024b). They use SCALE = 400, INIT_RATING = 1,000, and a further shift ANCHOR_SHIFT to produce the displayed scores:

$$\text{ELO}_i = \text{SCALE} \cdot \widehat{\theta}_i + \text{INIT\_RATING} + \text{ANCHOR\_SHIFT}.$$

The final constant (ANCHOR_SHIFT) shifts all the ELO$_i$ scores so that a specific reference model has a certain score. Chatbot Arena uses mixtral-8x7b-instruct-v0.1 as the reference model, assigning it a fixed score of 1,114. We use the same reference model in our analysis of Chatbot Arena; however, we assign the model a fixed score of 0 (a design choice that does not impact rankings). We note that the affine transformation does not affect model rankings since it is strictly monotonic and does not affect our proposed procedure since linear transformations can commute with first-order Taylor expansion.

**Setup for Data-Dropping.** We study whether dropping a small fraction $\alpha \in (0, 1)$ (e.g., $\alpha = 0.01$) of the preference data can change the ordering of the estimated BT scores. Broderick et al. (2020) define the *Maximum Influence Perturbation* as the largest possible change induced in a quantity of interest by removing at most $100\alpha\%$ of the data.

Let $w_n$ denote a weight on the $n$th data point, and collect these into a vector $w := (w_1, ..., w_N)$. Define the weighted estimator as

$$\hat{\boldsymbol{\theta}}(w) := \arg\max_{\boldsymbol{\theta}:\theta_1=0} \sum_{n=1}^{N} w_n \Big[ w_{WL} I_{y_n=W} \log \sigma(\theta_{i_n} - \theta_{j_n}) + w_{WL} I_{y_n=L} \log(1 - \sigma(\theta_{i_n} - \theta_{j_n}))$$
$$+ w_T I_{y_n=T} \left\{ \log \sigma(\theta_{i_n} - \theta_{j_n}) + \log(1 - \sigma(\theta_{i_n} - \theta_{j_n})) \right\} \Big]. \tag{4}$$

Setting $w = 1_N$ (the all-ones vector) recovers the BT scores computed on the full data (e.g., the original arena), while setting $w_n = 0$ corresponds to dropping the $n$th data point (e.g., a matchup). We define the set of all weight vectors corresponding to dropping at most an $\alpha$-fraction of the data as follows.

**Definition 1** (Feasible Drop Set). Let $W_\alpha := \{w \in \{0, 1\}^N : \sum_{n=1}^{N}(1 - w_n) \le \alpha N\}$ be the set of all binary weight vectors indicating subsets where at most $100\alpha\%$ of the data has been dropped.

---

[1]For ease of exposition in the main text, we assume there are no ties. In practice, we did not observe ties in Bradley–Terry-based point estimates. For rankings based on confidence intervals, we do observe and handle ties; see Appendix A.1 for details.

[2]"Chatbot Arena Leaderboard Calculation (Bradley–Terry model)" Colab notebook: https://colab.research.google.com/drive/1KdwokPjirkTmpO_P1WByFNFiqxWQquwH.

We begin by considering the ordering of BT scores between a pair of players, $i$ and $j$. Without loss of generality, we assume[3] that player $i$ has the higher estimated BT score on the full data:

$$\widehat{\theta}_i(1_N) \geq \widehat{\theta}_j(1_N).$$

We are interested in whether this ordering can be reversed by dropping at most an $\alpha$-fraction of the data.

We now extend this notion to an arena with $M$ players, for any $M \geq 2$. Let $\mathcal{T}(w) := \{\widehat{\theta}_i(w)\}_{i=1}^M$ denote the set of BT scores under weighting $w$. Let $\mathrm{rank}\left[\widehat{\theta}_i(w); \mathcal{T}(w)\right]$ denote the rank of a model under the weighting $w$.

**Definition 2** (Top-$k$ Set). We define the *top-$k$ set* under a data weighting $w$ as the set of players whose scores rank among the top $k$:

$$\mathcal{K}_{\mathcal{T}(w)} := \left\{i : \mathrm{rank}\left[\widehat{\theta}_i(w); \mathcal{T}(w)\right] \leq k\right\}. \tag{5}$$

**Definition 3** (Top-$k$ Data-Dropping Robustness). An arena is *top-$k$ robust at level $\alpha$* if no $\alpha$-fraction subset of data can be dropped to change the top-$k$ set for the full data. That is,

$$\left\{w \in W_\alpha : \mathcal{K}_{\mathcal{T}(1_N)} \neq \mathcal{K}_{\mathcal{T}(w)}\right\} = \emptyset. \tag{6}$$

Notice that Equation (6) is nontrivial to directly verify; to check directly, we could test out dropping all possible small-fraction subsets of the arena, but this combinatorial operation is computationally intractable in practice.

In Section 3, we show that verifying whether Equation (6) holds can be reduced to checking the robustness of a series of pairwise comparisons. Specifically, top-$k$ robustness as in Definition 3 can be checked by assessing whether there exists a reweighting $w \in W_\alpha$ that flips the ranking of a pair $(i, j)$ such that $i$ is inside and $j$ is outside the top-$k$ set. We then can test if such flipping can happen by: (a) using a continuous approximation of the discrete weights $w$ (also known as "approximate data-dropping") to identify a promising candidate subset of influential preferences, (b) dropping these, (c) recomputing the BT-based rankings, and (d) observing whether the rankings change. We detail this procedure in Section 3.

## 3 PROPOSED METHOD

Recall that our goal is to evaluate the robustness of the rankings induced by a BT-model when a small fraction of matches (e.g., evaluations) is removed from the arena. To this end, we introduce a method based on checking the robustness of pairwise BT score differences. We provide pseudocode for our method in Algorithm 1 and explain its steps below.

In Proposition B.1, we show that a top-$k$ set can be characterized by considering a set of pairwise comparisons. This result allows us to check top-$k$ robustness by checking pairwise robustness of all models inside the top-$k$ set against all models outside of this set. In the case that there does exist such a pair of models (one inside and one outside the top-$k$) whose rankings flip, then the top-$k$ set has changed, i.e., the arena is non-robust. In the case that there does not exist at least one such pair of models whose rankings can be flipped upon dropping a small fraction of preferences, then the top-$k$ set remains unchanged, i.e., the arena is top-$k$ robust.

Given the equivalence between checking the robustness of the top-$k$ set and checking the robustness of the aforementioned series of pairwise player comparisons, we propose a greedy algorithm to test whether the top-$k$ set is robust to worst-case data-dropping. Namely, we test the data-dropping robustness of all players in the top-$k$ set against all players outside of the top-$k$ set.

Before that, we describe what it means for a given pair of player scores, $(\widehat{\theta}_i(w), \widehat{\theta}_j(w))$, to be data-dropping robust. Without loss of generality, we assume throughout this section that player $i$ has the higher estimated BT score on the full data.

---

[3]If this assumption does not hold, the identities of $i$ and $j$ can be swapped.

**Pairwise Robust.** Given a pair of players, $(i, j)$, we say that the scores for this pair, $(\widehat{\theta}_i(w), \widehat{\theta}_j(w))$, are robust to small-fraction data-dropping at level-$\alpha$ if

$$\{w \in W_\alpha : \widehat{\theta}_i(w) < \widehat{\theta}_j(w)\} = \emptyset. \tag{7}$$

**Top-$k$ Robust.** Recall that an arena is top-$k$ robust at level-$\alpha$ if there does not exist a reweighting, $w \in W_\alpha$, such that $\mathcal{K}_{\mathcal{T}(1_N)} \neq \mathcal{K}_{\mathcal{T}(w)}$. Using the line of logic in Proposition B.1, this is equivalent to showing that, $\forall \, (i, j)$ where $i \in \mathcal{K}_{\mathcal{T}(w)}$ and $j \notin \mathcal{K}_{\mathcal{T}(w)}$, the pair $(\widehat{\theta}_i(w), \widehat{\theta}_j(w))$ is robust. Namely, if every comparison $(i, j)$ in this set of pairwise comparisons stays the same (after reweighting), then the top-$k$ set also stays the same (see Proposition B.1 for a detailed proof).

We now provide a method for checking the robustness of pairwise comparisons.

**Method for Checking Pairwise Robustness.** In Equation (7), we are interested in checking whether there exists a small fraction of evaluations, $w \in W_\alpha$, that can be dropped to change the sign of a difference in BT scores. Without loss of generality, we will assume that the sign of the difference of BT scores fit to the full data is positive (e.g., $[\widehat{\theta}_i(1_N) - \widehat{\theta}_j(1_N)] > 0$, meaning that model $i$ has a higher score than model $j$).

To evaluate the robustness of the sign of $[\widehat{\theta}_i(1_N) - \widehat{\theta}_j(1_N)]$ to dropping a small fraction of matches, we adopt a recently-developed method from the statistics literature known as the *Approximate Maximum Influence Perturbation* (Broderick et al., 2020) (see Appendix C.3 for a more detailed discussion on how we adapt this method to our problem setup). This method approximates the maximal directional change in a statistic, e.g., $[\widehat{\theta}_i(1_N) - \widehat{\theta}_j(1_N)]$, that can result from dropping a worst-case subset of data points (in our case, evaluations) of size at most $\lfloor \alpha N \rfloor$. This method allows us to sidestep running an expensive combinatorial search over all data subsets for the worst-case subset of matches to drop, a procedure that is computationally prohibitive for large LLM evaluation platforms like Chatbot Arena.

The optimization problem implied by the Maximum Influence Perturbation problem in our particular case is shown below,

$$\max_{w \in W_\alpha} \left( \left[ \widehat{\theta}_i(1_N) - \widehat{\theta}_j(1_N) \right] - \left[ \widehat{\theta}_i(w) - \widehat{\theta}_j(w) \right] \right). \tag{8}$$

We approximate this discrete optimization problem using AMIP approximation (Broderick et al., 2020); the idea is that, instead of solving the optimization directly, we first approximate the effect of dropping data by a first order Taylor expansion of the quantity $\widehat{\theta}_i(w) - \widehat{\theta}_j(w)$ over data weights $w$ and then solve the approximated optimization problem. In Appendix C, we provide a review of the general AMIP approximation, then formulate both the weighted and unweighted BT models as logistic regressions, and finally provide an explicit form of the approximation for both BT models.

Let the approximate solution to Equation (8) that is returned by AMIP be denoted as $\tilde{w}$ (i.e., the set of data weights that are 0 at indices of data points that AMIP chooses to drop and 1 elsewhere). For a candidate pair of players, $(i, j)$, we check whether after dropping, $[\widehat{\theta}_i(\tilde{w}) - \widehat{\theta}_j(\tilde{w})] < 0$. In other words, we refit the BT-model upon leaving out the subset of impactful evaluations identified by AMIP and check whether leaving out this subset induces a sign change in the difference of BT scores for the pair, $(i, j)$. We say that the BT scores for a pair of players, $(i, j)$, are non-robust if the *sign* of the difference in scores *becomes negative* upon refitting under $\tilde{w}$, (i.e., if $[\widehat{\theta}_i(\tilde{w}) - \widehat{\theta}_j(\tilde{w})] < 0$).

**Method for Checking Top-$k$ Robustness.** We now describe how we can fold our check for pairwise robustness into an overall routine for checking for top-$k$ robustness.

Recall from earlier in Section 3 that we can check top-$k$ robustness by checking pairwise robustness for every comparison $(i, j)$ where $i \in \mathcal{K}_{\mathcal{T}(w)}$ and $j \notin \mathcal{K}_{\mathcal{T}(w)}$. This amounts to checking the pairwise robustness for at most $k(M - k)$ pairs.

Thus, we check top-$k$ robustness by iterating over pairs of players. Note that, when checking the robustness of a given pair $(i, j)$, we allow matches between any two models (not only $(i, j)$) to be dropped. Since we only need to find one non-robust pair to render the set non-robust, not all pairs need to be checked. To save on compute, we take a greedy approach and start with comparing the most closely-ranked pairs between the top-$k$ ranked players and the remaining $M - k$ players, where

"closeness" is quantified using the absolute difference in BT scores fit on the full data;[4] pairs with smaller BT-score gaps are more likely to exhibit data-dropping non-robustness. Upon finding any single pair that is pairwise non-robust at an $\alpha$-level, the procedure terminates early and returns the corresponding players and the indices of the dropped evaluations. We say that an arena is $\alpha$-level top-$k$ robust if there does not exist a pair of players $(i, j)$, where $i \in \mathcal{K}_{\mathcal{T}(w)}$ and $j \notin \mathcal{K}_{\mathcal{T}(w)}$, that are $\alpha$-level pairwise non-robust. While our method uses an approximation to *identify* the influential preferences; it then performs an exact recomputation of the Bradley–Terry scores with the identified preferences removed. As a result, all non-robustness reported in this paper is definitive: when we state that dropping $100\alpha\%$ of preferences changes the ranking, we have explicitly verified that the ranking does in fact change upon removal of the surfaced subset. However, the algorithm may not catch all cases of non-robustness (i.e., false negatives are possible). See Appendix H for an extended discussion on the possibility of false negatives.

**Runtime.** The above procedure is fast for assessing the robustness of preference-based ranking systems. For example, we tested our method on historical preference datasets released by the Chatbot Arena project and hosted on Hugging Face (Chiang et al., 2024a). Specifically, we check top-1 and top-5 robustness on a dataset of size around $50,000$ evaluations in under 3 minutes on a personal computer equipped with an Apple M1 Pro CPU at 3200 MHz and 16 GB of RAM.

# 4 EXPERIMENTS

Our analysis reveals that 1) dropping as little as $0.003\%$ of the evaluation data can flip the top-ranked model in popular LLM evaluation platforms (Section 4.2), 2) crowdsourced human-evaluated systems are about as non-robust as AI-evaluated systems (Section 4.3), 3) the LLM-generated responses of the dropped evaluations appear similar in content (Section 4.4), and 4) sensitivity depends on BT score margins (Appendix F.1). Henceforth, for convenience, we use "robustness" as shorthand for robustness of a system's top-$k$ ranking to dropping a small fraction, $\alpha$, of the data.

## 4.1 DATA AND SETUP

We run our robustness check on a variety of LLM Arenas, including Chatbot Arena (Chiang et al., 2024a), MT-bench (Zheng et al., 2023), Search Arena (Miroyan et al., 2026), Webdev Arena (Vichare et al., 2025), and Vision Arena (Chou et al., 2025). For more information about each arena, see Appendix D. Our analysis relies on historical preference datasets released by the Chatbot Arena project (Chiang et al., 2024a) and publicly hosted on LMArena's HuggingFace account. Each record represents a matchup consisting of two LLMs that answer the same prompt, the names of the two models, and the user label indicating preference for model A, model B, or a tie. Figure 3 presents the Bradley–Terry scores of the top-10 models on Chatbot Arena.

To compare the robustness of LLM arenas to more classical use cases of BT models, we also run our check on two sports datasets, namely NBA (FiveThirtyEight, 2025) and ATP tennis (Sackmann, 2024). For details on the sports datasets, see Appendix D.

For each dataset, we assess top-$k$ robustness with $k \in \{1, 3, 5, 10, 20\}$, extending up to the maximum number of models present in the respective arena when fewer than 20 models are present.

## 4.2 SENSITIVITY OF LLM ARENAS

We find many popular LLM arenas to be incredibly sensitive to data-dropping (see Table 1). In particular, we find that dropping just two ($0.003\%$ of) evaluations is enough to change the top-ranked model on Chatbot Arena from GPT-4-0125-preview to GPT-4-1106-preview; see the two surfaced prompts and response pairs in Appendix F. We then find that dropping just three ($0.005\%$ of) evaluations can change one of the models in the top-5 rankings (the 5th and 6th-ranked models changed). Surprisingly, GPT-4-1106-preview participated in the most matchups across the entire arena and GPT-4-0125-preview also participated in a sizable number of matchups, as shown in

---

[4]The robustness of the relative ranking of two players is correlated with the proximity of their BT scores as seen in Figure 18.

---

**Algorithm 1** Our Data-dropping Robustness Check on Rankings

---

1: **Input:** Dataset $\mathcal{D}$ (the collection of matches, e.g., preferences), rank $k$, drop fraction $\alpha$.
2: **Output:** (1) A determination of whether top-$k$ non-robustness was found. (2) If top-$k$ non-robustness is found, we additionally return the pair of players $(i, j)$ whose rankings flipped, the differences in their scores pre- and post- data-dropping, and the most influential set $I$.
3:
4: ▷ **Fit a Bradley–Terry model on the full arena.**
5: $\hat{\boldsymbol{\theta}}(1_N) \leftarrow \text{FitBTModel}(\mathcal{D})$
6: Determine the top-$k$ set, $\mathcal{K}_{\mathcal{T}(1_N)}$, from $\hat{\boldsymbol{\theta}}(1_N)$.
7:
8: ▷ **Compute score gap for each player pair of interest.**
9: $P \leftarrow \{(i, j) : i \in \mathcal{K}_{\mathcal{T}(1_N)}, j \notin \mathcal{K}_{\mathcal{T}(1_N)}\}$
10: **for** each $(i, j) \in P$ **do**
11:     Compute score gap $\hat{\Delta}(1_N)_{ij} \leftarrow |\hat{\theta}_i(1_N) - \hat{\theta}_j(1_N)|$
12: **end for**
13: Sort pairs $(i, j)$ in $P$ by increasing $\Delta(1_N)_{ij}$
14:
15: ▷ **Check pairwise robustness by choosing pairs in order of increasing score gap.**
16: **for** each $(i, j)$ in sorted $P$ **do**
17:     ▷ **Compute influence scores.**
18:     **for** each datapoint (preference) $n$ **do**
19:         $\text{IF}_n(i) \leftarrow$ influence score for datapoint $n$ on $\hat{\theta}_i(1_N)$
20:         $\text{IF}_n(j) \leftarrow$ influence score for datapoint $n$ on $\hat{\theta}_j(1_N)$
21:         $\Delta_n(i, j) \leftarrow \text{IF}_n(i) - \text{IF}_n(j)$
22:     **end for**
23:
24:     ▷ **Identify worst-case subset by sorting influence scores.**
25:     Choose the $\lfloor \alpha N \rfloor$ values of $\Delta_n(i, j)$ that are the largest in the negative direction, assuming that $\hat{\theta}_i(1_N) - \hat{\theta}_j(1_N) > 0$.
26:     $I \leftarrow$ indices corresponding to the $\lfloor \alpha N \rfloor$ most negative $\Delta_n$ values.
27:
28:     ▷ **Compute the AMIP-predicted score difference.**
29:     $(\hat{\theta}_i(w) - \hat{\theta}_j(w))_{\text{AMIP}} \leftarrow (\hat{\theta}_i(1_N) - \hat{\theta}_j(1_N)) + \sum_{n \in I} \Delta_n$
30:
31:     ▷ **Compute the exact refit for verification.**
32:     $\hat{\boldsymbol{\theta}}(w) \leftarrow \text{FitBTModel}(\mathcal{D}_{-\alpha})$, where $\mathcal{D}_{-\alpha}$ is the set of points in $\mathcal{D}$ except those with indices in $I$
33:     Compute new difference: $(\hat{\theta}_i(w) - \hat{\theta}_j(w))$
34:
35:     **if** $\text{sign}((\hat{\theta}_i(1_N) - \hat{\theta}_j(1_N))) \neq \text{sign}(\hat{\theta}_i(w) - \hat{\theta}_j(w))$ **then**
36:         **return** "Arena is $\alpha$-level top-$k$ non-robust", $(i, j), (\hat{\theta}_i(1_N) - \hat{\theta}_j(1_N)), (\hat{\theta}_i(w) - \hat{\theta}_j(w)), I$
37:     **end if**
38: **end for**
39:
40: **return** "Arena was not found to be $\alpha$-level top-$k$ non-robust"

---

Figure 17, suggesting that data-dropping sensitivity cannot be attributed to a small sample size alone.

In addition to reporting rankings based on point-estimate BT-scores, LMArena reports an approximate ranking based on the end points of bootstrap confidence intervals (see LMArena (2025); Chiang et al. (2024b;a)). Even with the bootstrap-based rankings, we still find arenas to be surprisingly sensitive to worst-case data-dropping. For instance, we surface arenas where the bootstrap-based ranking outputs a single top-ranked model, but upon small-fraction data dropping, the model becomes no longer the sole top-ranked model (see Figure 7 in Appendix A.1). See Table 2 in Appendix A.1 for more details on the sensitivity of LMArena rankings based on bootstrap confidence intervals.

Out of the LLM arenas we analyze, MT-bench is the sole benchmark that is robust at an $\alpha$-level of 0.01 (see Table 1). Here, dropping 92 out of 3,355 (2.74% of) evaluations changes the top model from GPT-4 to Claude-v1. Dropping 110 (3.28% of) matchups can change one of the models in the top-5 rankings (again, the 5th and 6th ranked models changed). There are several reasons that

| Arena | Evaluator (Judge) | Number Dropped | Percentage Dropped |
|---|---|---|---|
| Chatbot Arena | Human | 2 out of 57477 | 0.00348% |
| Vision Arena | Human | 28 out of 29845 | 0.0938% |
| NBA Games | NA | 17 out of 109892 | 0.0155% |
| Chatbot Arena | LLM | 9 out of 49938 | 0.0180% |
| Webdev Arena | Human | 18 out of 10501 | 0.171% |
| Search Arena | Human | 61 out of 24469 | 0.253% |
| MT-bench | LLM | 40 out of 2400 | 1.67% |
| ATP Tennis | NA | 6 out of 278 | 2.16% |
| MT-bench | Human | 92 out of 3355 | 2.74% |

Table 1: Results of checking top-1 robustness of BT-scores on each of the arenas, listed in ascending order of robustness (from the least to the most robust). The "Number Dropped" column reports the number of preferences (matches) that are sufficient to flip the first and second-place models (players). The "Percentage Dropped" column shows this number as a percentage of the number of total preferences in the full arena. Datasets we found to be robust at an $\alpha$-level of $1\%$ are colored in gray.

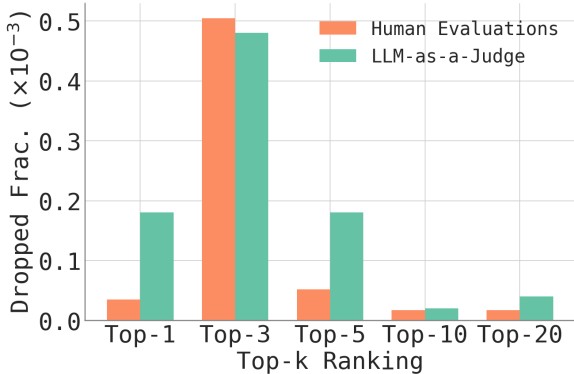

Figure 2: Each bar shows the fraction of data points dropped from Chatbot Arena that is sufficient to demote the BT score of a model inside the top-$k$ to outside of the top-$k$ ($k \in \{1, 3, 5, 10, 20\}$). The orange bars correspond to human evaluators and green bars to LLM-as-a-judge evaluators.

may lead MT-bench to be much more robust than the other LLM arenas. MT-bench consists of 80 carefully-designed multi-turn questions intended to differentiate models on core capabilities such as math, reasoning, and writing, and annotated by expert annotators (Zheng et al., 2023). In contrast, all other arenas in our analysis are large-scale crowdsourced platforms, which rely on user-submitted prompts and crowd-sourced preference judgments.

### 4.3 HUMANS VS. LLM-AS-A-JUDGE

Within arenas that used both human and LLM judges, we find neither human-annotated nor LLM-annotated datasets to be clearly more sensitive than the other to worst-case data-dropping (see Table 1 and Figure 2). For Chatbot Arena, we find that the human-annotated dataset is slightly more sensitive (required dropping fewer evaluations) for $k \in \{1, 5, 10, 20\}$ while the LLM-annotated dataset is slightly more sensitive for $k = 3$ (see Figure 2). In contrast, for MT-bench, the LLM-annotated dataset is more sensitive than the human-annotated dataset for all $k \in \{1, 3, 5\}$, perhaps due to the use of expert-human annotators.[5] Taken together, we cannot conclude that rankings based on human preferences nor those based on LLM-as-a-judge preferences are systematically more sensitive than the other.

### 4.4 INSPECTING DROPPED PREFERENCES

Our method can identify the prompts and response-pairs responsible for changing top leaderboard rankings. On Chatbot Arena, we find that dropping just *two* human evaluations suffices to flip the

---

[5]We do not test $k \in 10, 20$, as MT-bench includes only six models.

rankings of GPT-4-1106-preview (originally ranked first) and GPT-4-0125-preview (ranked second). We provide these prompts and response pairs in Appendix F. A qualitative analysis of the prompt-response pairs (see Appendix F) shows that the two surfaced preferences correspond to cases that a strong judge model (GPT-5.1) identifies as atypical (i.e., different to what the "typical" user might prefer). In both cases, GPT-4-1106-preview was judged to have lost against substantially lower-ranked models: Vicuna-13b (ranked 43rd) and Stripedhyena-nous-7b (ranked 45th). Dropping these two anomalous losses is enough to raise GPT-4-1106-preview's position from second to first.

## 5 RELATED WORK

### 5.1 VULNERABILITIES IN AI LEADERBOARDS

Despite its ease-of-use and widespread popularity, large-scale, community-driven platforms like Chatbot Arena are found to be vulnerable to adversarial attacks that can distort model rankings. Min et al. (2025) demonstrate that Chatbot Arena is vulnerable to vote-rigging: by injecting just a few hundred manipulated votes (out of 1.7 million), attackers can significantly change the top model rankings. Similarly, Huang et al. (2025b) find that an attacker can accurately identify which model produced a response on Chatbot Arena, and use that to systematically upvote or downvote a target model. They propose several defenses (e.g., authentication, rate limits, malicious-vote detection) that make the leaderboard more robust to adversarial agents. Injected votes may be especially easy to construct on LLM-as-a-judge systems, as recent works show that LLM judges can be gamed in systematic ways (Zheng et al., 2025; Raina et al., 2024). Beyond vote-rigging, Singh et al. (2025) identify other issues such as data leakage and private testing practices that allow large, proprietary model developers to selectively report the best-performing versions of their models on the arena. Zhao et al. (2025) present a case study showing that model rankings can shift when a fraction of votes comes from apathetic or arbitrary annotators. Their analysis finds that replacing $10\%$ of votes with uniform $\{0, 1\}$ labels can move two models by up to five ranks. In contrast, we do not alter votes but instead demonstrate that rankings can change by removing an alarmingly small fraction ($0.003\%$) of the votes. More importantly, while Zhao et al. (2025) present a case study focused on the rankings of three specific test models, we develop a systematic method to evaluate the robustness of BT-based ranking systems under worst-case data dropping, which also identifies the specific prompt–response pairs driving ranking flips. Beyond pairwise-preference-based rankings, past works have pointed out the fragility of LLM ranking systems based on absolute performance scores. Perlitz et al. (2024) show that rankings derived from the Mean Win Rate (MWR) metric (a variant of the Borda count) can be gamed to increase the ranking of a target model by evaluating numerous models that are almost equal to, but slightly weaker than, the target model, thereby inflating the target model's ranking. The authors suggest allocating more evaluation resources for models ranked at the top of the leaderboard, where the rankings may carry more weight. Finally, while all pairwise-preference-based related works in this section focus on Chatbot Arena, we extend our analysis to other domains (vision, web design, search, and multi-turn dialogue) and find the leaderboard rankings on these platforms to be similarly non-robust.

### 5.2 DATA-DROPPING ROBUSTNESS

A growing body of works in statistics and theoretical computer science develops algorithms for assessing whether data analyses are robust to dropping a small, worst-case fraction of the data (Broderick et al., 2020; Kuschnig et al., 2021; Moitra & Rohatgi, 2023; Freund & Hopkins, 2023; Nguyen et al., 2024; Huang et al., 2025a; Rubinstein & Hopkins, 2025). To our knowledge, only one prior work has investigated this question in the context of ranking systems: Shiffman et al. (2023) study the robustness of rankings in gene set enrichment analysis, showing that dropping just a few cells can alter the ranking of p-values derived from the hypergeometric test. In contrast, our work examines ranking robustness in a BT-based ranking system. While Shiffman et al. (2023) analyze p-value rankings, we analyze preference-based rankings of LLMs, extending AMIP (Broderick et al., 2020) to study the robustness of BT-based ranking systems.

## 6 DISCUSSION

Crowdsourced LLM evaluation platforms like Chatbot Arena offer a way to rank LLMs by aggregating preferences over responses to open-ended prompts. There is good reason that this setup has been widely-adopted: it is easy to scale, doesn't require expert annotators, and enables the aggregation of many prompts and judgments across a wide range of users (Zheng et al., 2023; Don-Yehiya et al., 2025).

In theory, this aggregation helps average out individual annotator variability and yields a signal that is generalizable. However, in practice, we find that model rankings can depend on just a small handful of human (or LLM) evaluations. Thus, we encourage users of leaderboards and benchmark contests to run our method to investigate the fragility of crowdsourced LLM evaluation platforms before publishing results.

Sensitivity to worst-case data-dropping is often indicative of low signal-to-noise in the underlying data (Broderick et al., 2020); to help increase signal-to-noise, we recommend three different design-related improvements that AI arenas could make. (1) Collect richer forms of feedback beyond binary preferences (e.g., asking for evaluators' confidence levels (Méndez et al., 2022).[6] (2) Design more discriminative prompts. Arenas could incorporate a prompt-filtering system to identify and remove uninformative prompts, or create tools to identify prompts requiring specialized knowledge in order to route them to appropriate evaluators (Don-Yehiya et al., 2025). Chiang et al. (2024a) perform topic-modeling of the prompts submitted to Chatbot Arena. Their top-16 topics include "Poetry Writing Prompts" and "Movie Recommendations and Ratings." The subjective nature of such topics may make differentiation between top models less meaningful. (3) Ensure higher-quality preference annotations. Arenas could use mediators to perform fine-grained assessments of crowdsourced responses (Don-Yehiya et al., 2025), and categorize prompts by instruction type (e.g., factual recall, creative generation) to promote more fine-grained model comparisons within categories (Chia et al., 2024).

A complementary line of work on creating high-quality synthetic benchmarks argues that separability—requiring performance gaps between models to be wide enough for leaderboard trends to remain stable under subsampling—should be a main design criterion (Li et al., 2025). At the same time, our findings may suggest that apparent leaderboard differences may be artifacts of noise in the evaluation process rather than genuine performance gaps, which cautions against treating AI leaderboard rankings as definitive indicators of differences in model performance.

ACKNOWLEDGMENTS

This work was supported in part by an ONR Early Career Grant, the MIT-IBM Watson AI Lab, the NSF TRIPODS program (award DMS-2022448), a MachineLearningApplications@CSAIL Seed Award, and the Amazon AI Research Innovation Fellowship. We thank Hao Sun from the University of Cambridge and Google Deepmind for helpful initial discussions and for pointers to references.

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

# APPENDIX

## A  UNCERTAINTY QUANTIFICATION

### A.1  SENSITIVITY OF LLM ARENA RANKINGS BASED ON BOOTSTRAP CONFIDENCE INTERVALS

In addition to reporting rankings based on point-estimate BT-scores, LMArena reports an approximate ranking based on the end points of bootstrap confidence intervals (see LMArena (2025); Chiang et al. (2024b;a)). Specifically, Chiang et al. (2024a) computes bootstrap-confidence-interval-based rankings, which we will henceforth refer to as *bootstrap-based rankings*, as

$$R_m = 1 + \sum_{m' \in [M]} \mathbf{1}\{\inf C_{m'} > \sup C_m\},\tag{9}$$

where $R_m$ denotes the rank and $C_m$ the bootstrap confidence interval of model $m$. Under this scheme, a model's ranking increases by one for every other model whose lower confidence-interval endpoint exceeds the upper endpoint of the model in question (see Equation (9)). In other words, a model, $m$, is ranked below all models whose performance is significantly higher according to non-overlapping bootstrap confidence intervals. This definition (see Equation (9)) induces a set-valued ranking: multiple models may share the same ranking whenever their confidence intervals overlap with one another. Thus, a bootstrap-based "rank" corresponds often to a set of statistically indistinguishable models, rather than a single model.

In the bootstrap-based ranking setting, we follow the same notion of top-$k$ robustness introduced in Definition 3. An arena is deemed top-$k$ robust at level-$\alpha$ if no $\alpha$-fraction subset of data can be dropped to change the top-$k$ set of models. The only modification under the bootstrap-based ranking scheme is that each "rank" now corresponds to a set of statistically indistinguishable models. Thus, we regard the top-$k$ set as having changed whenever any model is added to or removed from this set.

| Arena | Evaluator (Judge) | Number Dropped | Percentage Dropped |
|---|---|---|---|
| Chatbot Arena | Human | 29 out of 57477 | 0.0510% |
| Search Arena | Human | 25 out of 24469 | 0.103% |
| Chatbot Arena | LLM | 75 out of 49938 | 0.150% |
| Vision Arena | Human | 125 out of 29845 | 0.419% |
| Webdev Arena | Human | 160 out of 10501 | 1.52% |
| MT-bench | Human | 92 out of 3355 | 2.74% |
| MT-bench | LLM | 40 out of 2400 | 4.00% |

Table 2: Results of checking top-1 robustness of bootstrap-based rankings on each of the arenas, listed in ascending order of robustness (from the least to the most robust). The "Number Dropped" column reports the number of preferences (matches) that are sufficient to flip the first and second-place models (players). The "Percentage Dropped" column shows this number as a percentage of the number of total preferences in the full arena.

To construct Table 2, we first compute the bootstrap-based rankings on the full dataset, apply our method to identify influential preferences, remove those preferences, and then recompute the

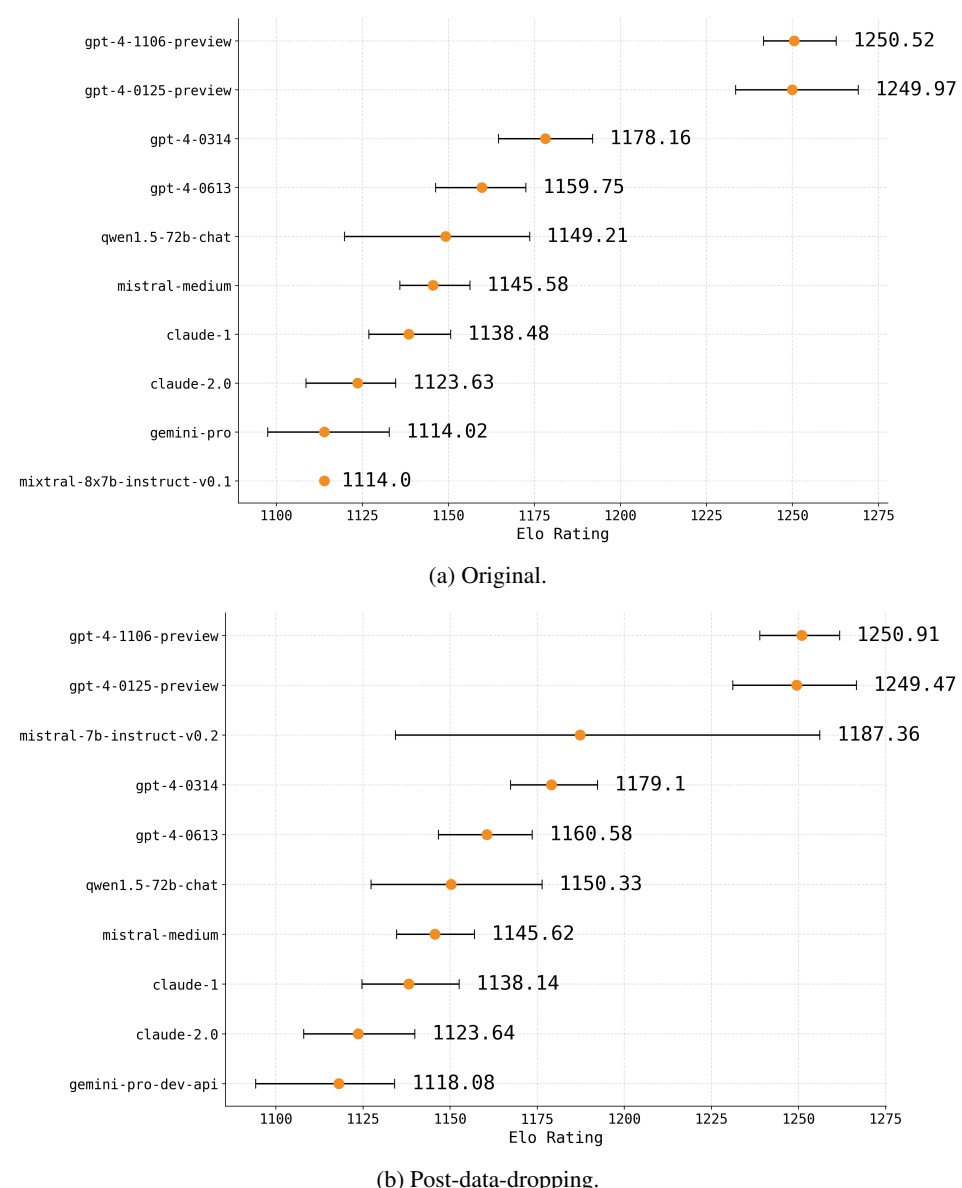

(a) Original.

(b) Post-data-dropping.

Figure 3: 95% Bootstrap-confidence-interval-based rankings on Chatbot Arena (Human Judge).

bootstrap-based rankings. Along with Table 2, we display the plots of the bootstrap-based rankings for the full data and the rankings post-data-dropping in Figures 3 to 9.

Despite the bootstrap's attempt to account for sampling uncertainty, we continue to find many arenas to be surprisingly sensitive to worst-case small-fraction data-dropping: the set of models ranked top-1 still changes in many arenas after removing a very small fraction of the arena. Across these experiments, we observe several arenas in which a new model enters the top-1 set (Figures 3, 5 and 6) and one arena in which a model is removed from the top-1 set (Figure 4), all from dropping less than 1% of preferences on the arena. We also surface arenas where the bootstrap-based ranking outputs a single top-ranked model, but upon small-fraction data dropping, the model becomes no longer the sole top-ranked model (see Figure 7).

This result shows that AMIP-based non-robustness is not an artifact of ignoring statistical uncertainty captured by confidence intervals. Rather, even after incorporating bootstrap variability, the arenas continue to be AMIP sensitive.

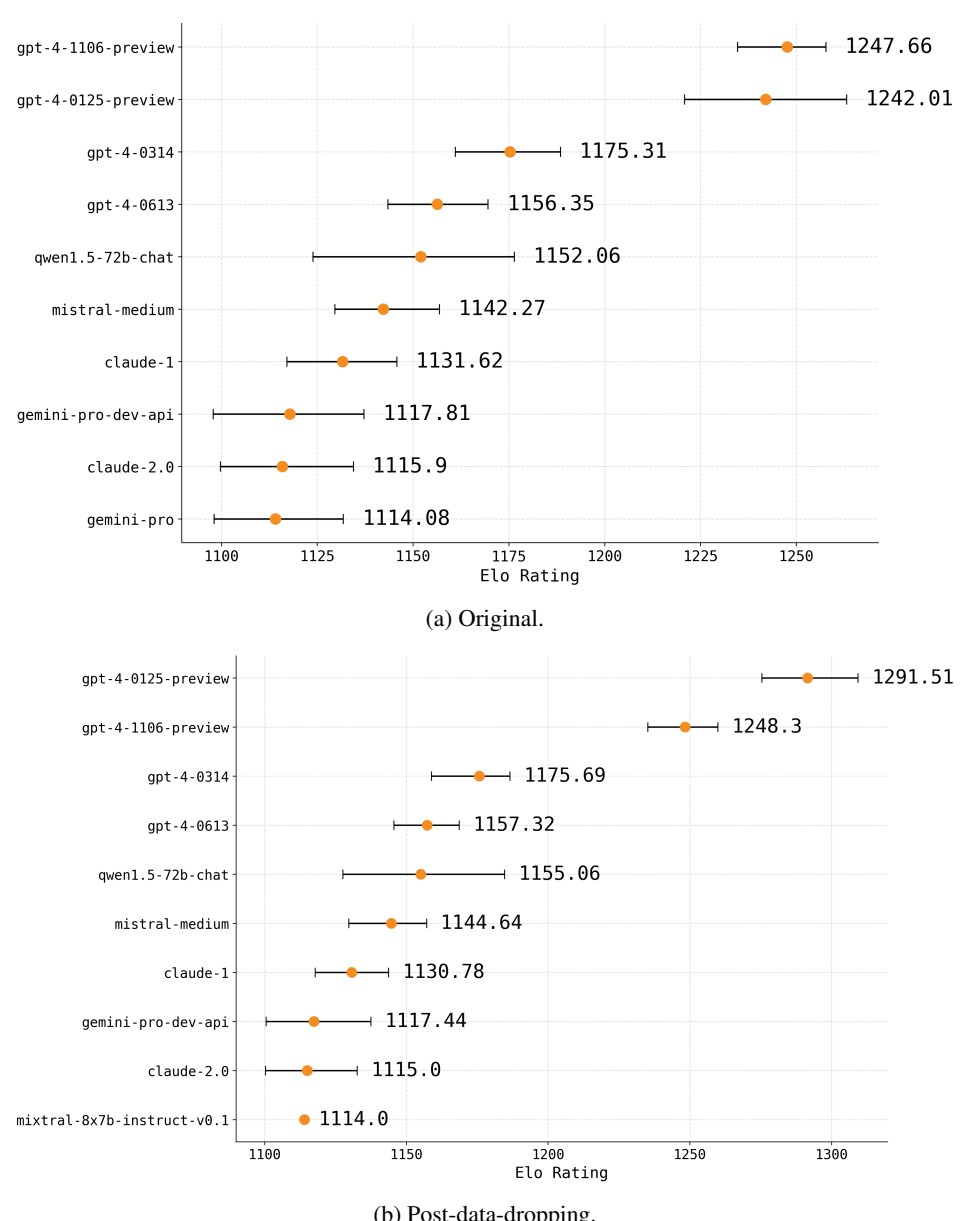

(a) Original.

(b) Post-data-dropping.

Figure 4: 95% Bootstrap-confidence-interval-based rankings on Chatbot Arena (LLM Judge).

### A.2 DISTINCTION BETWEEN WORST-CASE DATA-DROPPING SENSITIVITY AND CONFIDENCE INTERVALS

Confidence intervals, such as the bootstrap intervals reported on LMArena (LMArena, 2025), do quantify a form of sensitivity of BT-estimated rankings to variability across samples. However, the sampling-based sensitivity that bootstrap confidence intervals capture is conceptually different from that captured by AMIP. Bootstrap intervals characterize how much an estimate (e.g., the BT score) varies when data are resampled uniformly at random. In contrast, AMIP measures the maximum change in a BT-score difference that can be induced by removing a worst-case small fraction of the data. While frequentist (Gao et al., 2023; Hunter, 2004) confidence interval methods are meant to capture randomness in the data-generating process, the AMIP targets sensitivity of a single, fixed dataset. This focus on a single sample differs in spirit from the variability across "counterfactual worlds" that uncertainty quantification methods are meant to measure. In this sense, the two approaches answer complementary questions about the stability of a sample-based conclusion: the

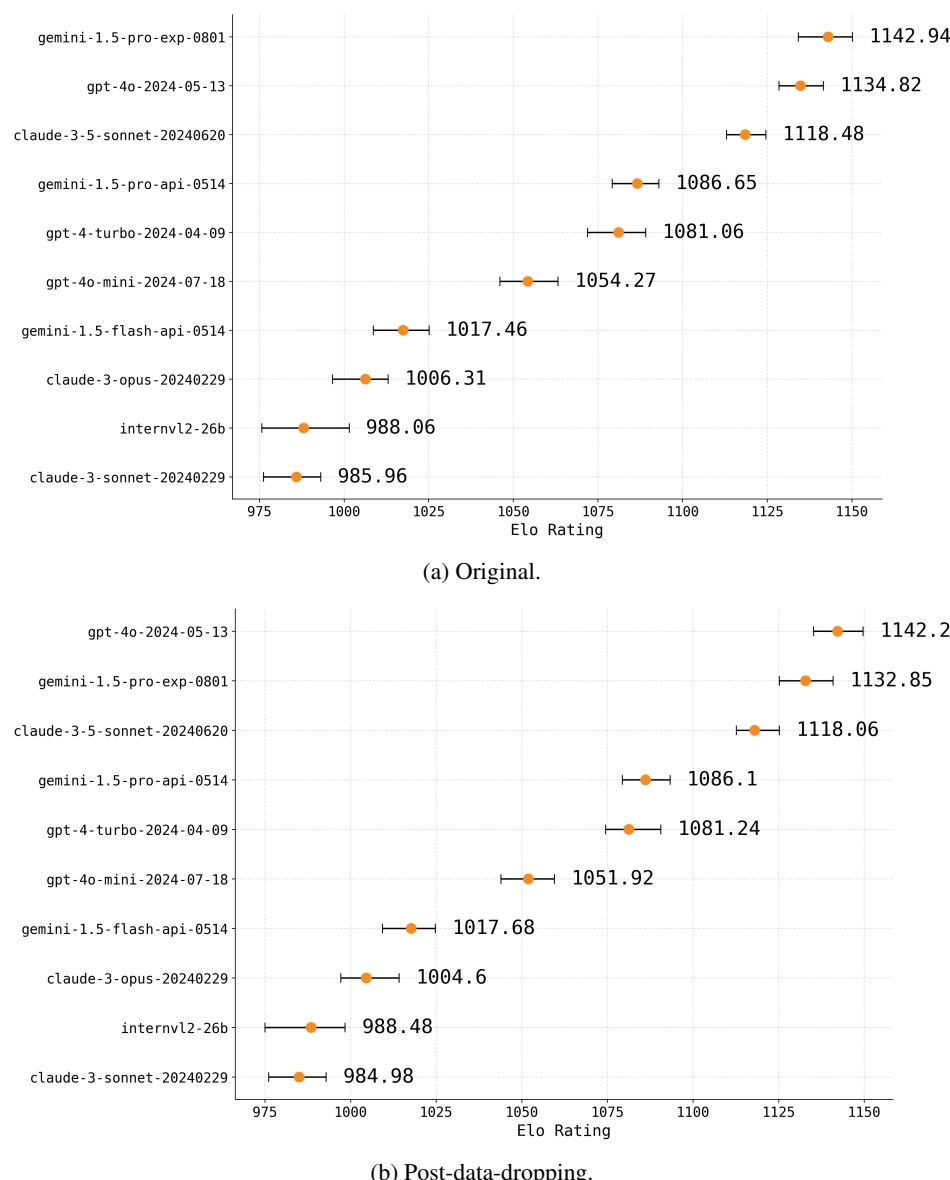

(a) Original.

(b) Post-data-dropping.

Figure 5: 95% Bootstrap-confidence-interval-based rankings on Vision Arena.

confidence intervals measure sampling uncertainty, while worst-case data-dropping robustness examines whether the conclusion is driven by a very small fraction of the observations in the sample. Although Bayesian credible intervals (Leonard, 1977) also operate under the case of a single, fixed dataset, past work has demonstrated that data analyses can be both statistically significant in the Bayesian sense (credible interval does not include zero) and still sensitive to worst-case data dropping (see Bayesian hierarchical model case study in Section 4.4 of (Broderick et al., 2020)). So, analogous to the frequentist case, the AMIP again represents a different and complementary check.

These tools also differ in the statistical assumptions under which they provide guarantees. Bootstrap-based confidence intervals rely on the data being i.i.d. draws from a target population. Real-world preference datasets often depart from this regime due to differences in annotators (e.g., the same, or similar types of, annotators may annotate several prompts on LMArena), resulting prompt-selection biases, and various other potential context-based factors. AMIP, by contrast, does not require an i.i.d. assumption and therefore remains valid in settings where classical resampling tools do not apply reliably. Prior work (Broderick et al., 2020) has demonstrated that data analyses can be si-

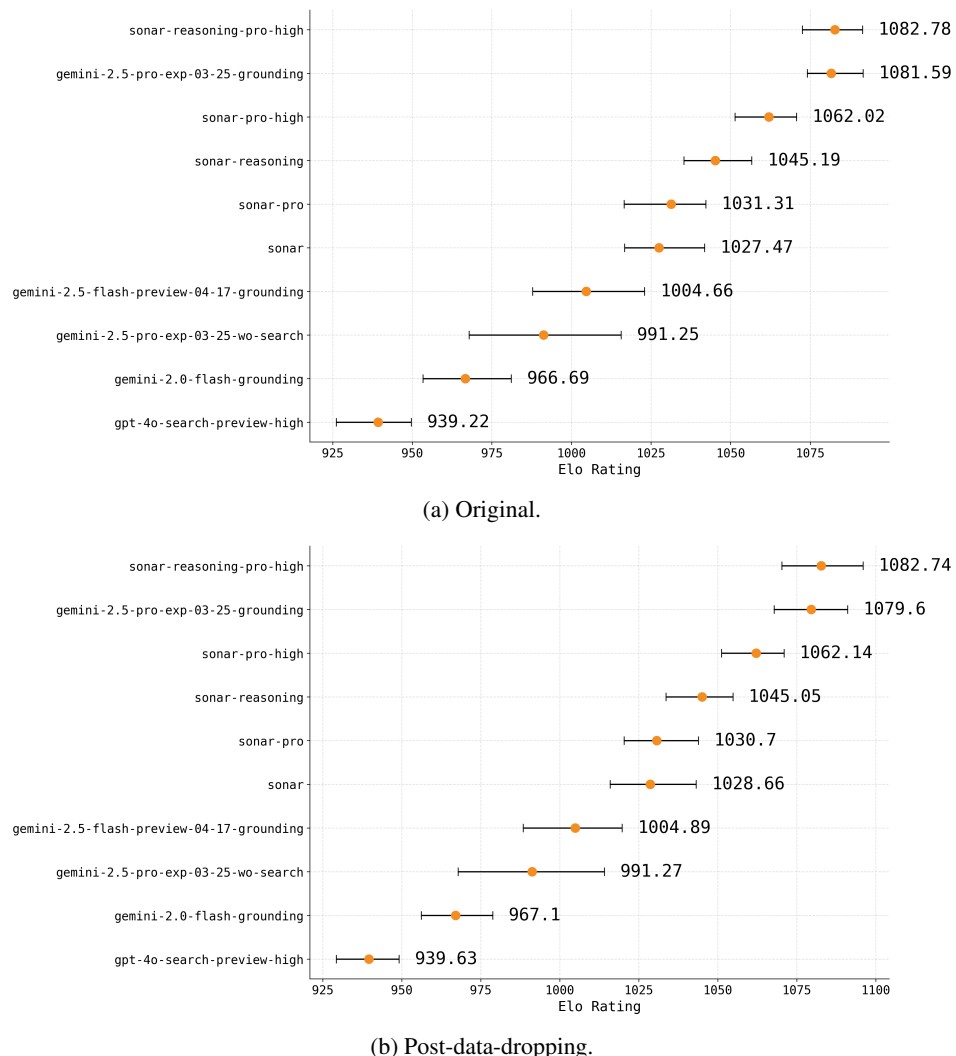

(a) Original.

(b) Post-data-dropping.

Figure 6: 95% Bootstrap-confidence-interval-based rankings on Search Arena.

multaneously statistically significant yet worst-case data-dropping non-robust. In this sense, AMIP provides a complementary and practically useful lens for assessing the generalizability of LLM leaderboard rankings.

## A.3    UNIFORM DATA-DROPPING EXPERIMENT

To examine the contrast between worst-case data-dropping and dropping random pairwise preferences, we conduct a uniform subsampling experiment. For each arena, we drop $1\%$ of the evaluations uniformly at random, repeat the experiment 100 times, and record the fraction of runs in which the top-ranked model remains unchanged relative to the full arena. For Chatbot Arena (human-judge), we additionally report robustness at a finer scale of $\alpha = 0.1\%$.

The results in Table 3 highlight a key conceptual distinction between uniform and worst-case data-dropping. Across nearly all arenas, dropping $1\%$ of the evaluations uniformly at random leaves the top-ranked model unchanged in every trial. Even Chatbot Arena (human-judge), which is the least stable under uniform subsampling, maintains its top-ranked model in $77\%$ of random $1\%$ deletions, a fraction that is many magnitudes larger than the $0.00348\%$ of preferences required to flip the top-ranked model when dropping the worst-case data subset. These results show that the rankings are extremely sensitive to dropping a worst-case small fraction of preferences, yet stable (at $\alpha = 1\%$) to

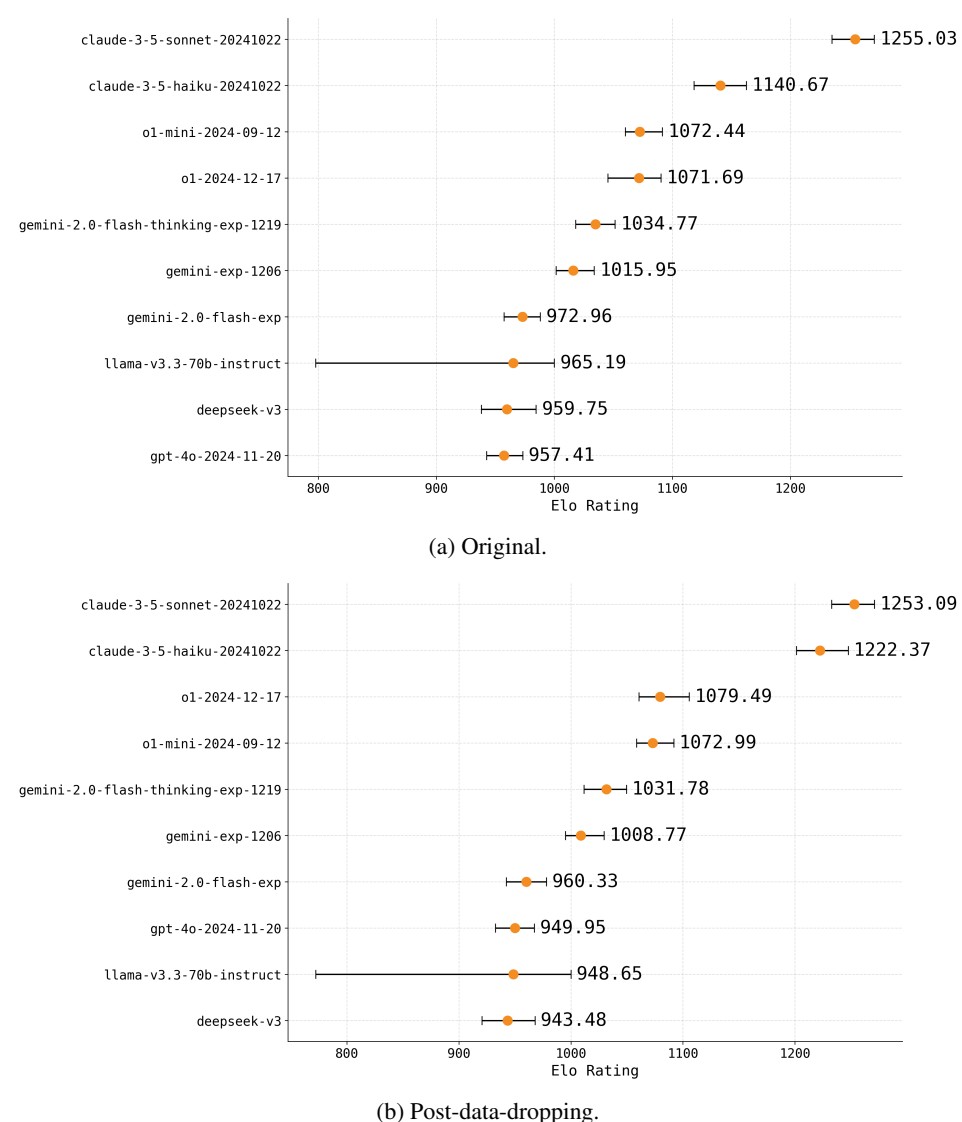

(a) Original.

(b) Post-data-dropping.

Figure 7: 95% Bootstrap-confidence-interval-based rankings on Webdev Arena.

dropping preferences chosen at random. Taken together, these observations show that uniform and worst-case data-dropping probe fundamentally distinct failure modes.

## B TOP-k SETS CAN BE CHARACTERIZED BY SETS OF PAIRWISE PLAYER COMPARISONS

We show in Proposition B.1 that the top-$k$ set can be characterized by a set of pairwise player comparisons.

**Proposition B.1.** *Suppose we have $M$ real numbers, $\mathcal{T}(w) := \{\widehat{\theta}_i(w)\}_{i=1}^M$. Suppose a set $\mathcal{S} \subset \mathcal{T}(w)$ satisfies $|\mathcal{S}| = k$. Suppose it is the case that $\forall\, \widehat{\theta}_i(w) \in \mathcal{S}$ and $\forall\, \widehat{\theta}_j(w) \in \mathcal{T}(w) \setminus \mathcal{S}$, we have that $\widehat{\theta}_i(w) > \widehat{\theta}_j(w)$. Then, it must be that $\mathcal{S}$ is the top-k set, i.e., $\mathcal{S} = \mathcal{K}_{\mathcal{T}(w)}$.*

*Proof.* We first show that $\mathcal{S} \subset \mathcal{K}_{\mathcal{T}(w)}$. Suppose that $\widehat{\theta}_i(w) \in \mathcal{S}$. By assumption, we have that $\forall\, \widehat{\theta}_j(w) \in \mathcal{T}(w) \setminus \mathcal{S}$, $\widehat{\theta}_i(w) > \widehat{\theta}_j(w)$. Since $|\mathcal{T}(w) \setminus \mathcal{S}| = M - k$, there must exist at least

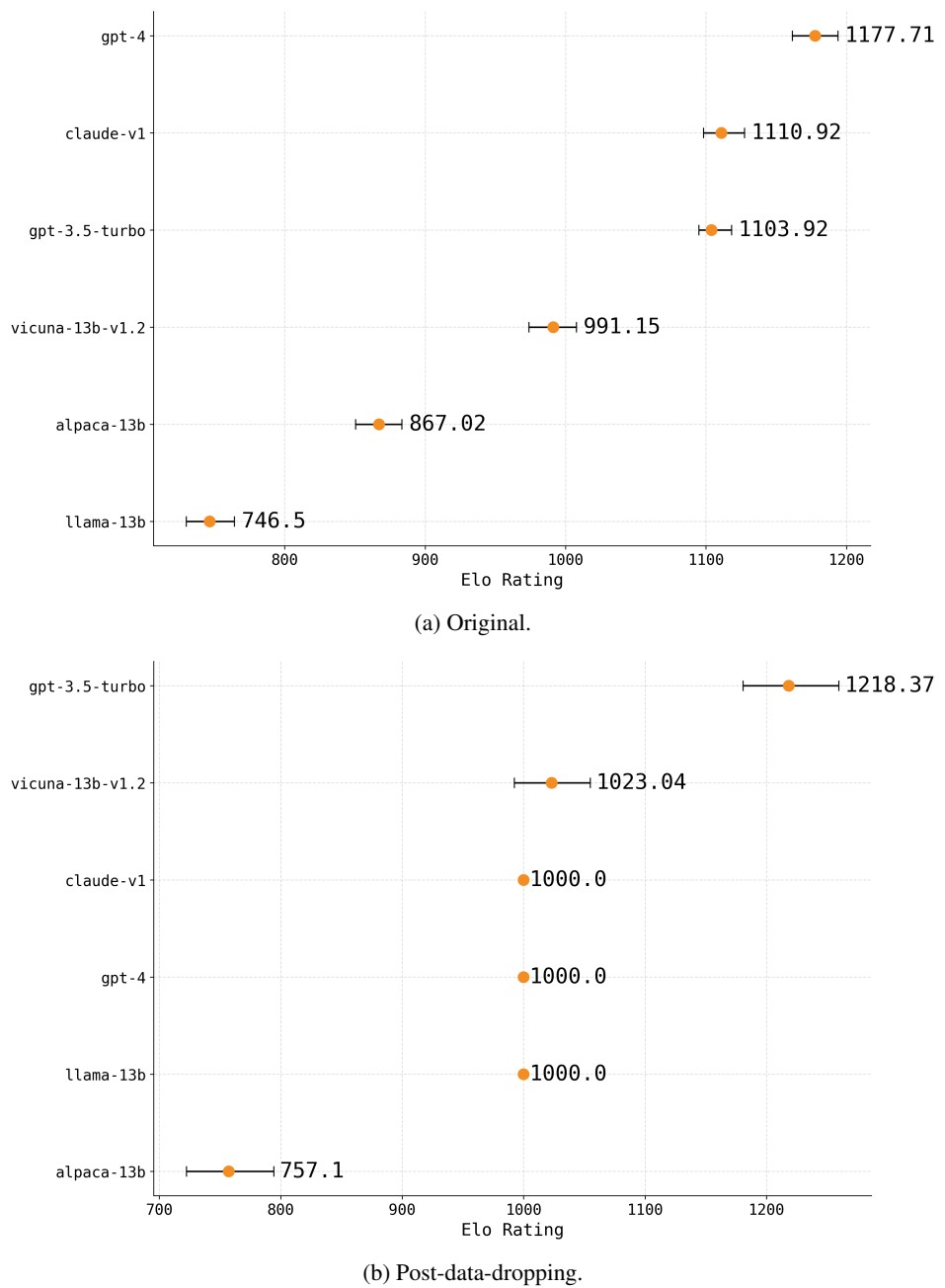

(a) Original.

(b) Post-data-dropping.

Figure 8: 95% Bootstrap-confidence-interval-based rankings on MTBench (Human Judge).

$(M - k)$ values in $\mathcal{T}(w)$ that are smaller than $\widehat{\theta}_i(w)$. This must mean that $\mathrm{rank}(\widehat{\theta}_i(w); \mathcal{T}(w)) \leq k$, so $\widehat{\theta}_i(w) \in \mathcal{K}_{\mathcal{T}(w)}$ as needed.

We next show that $\mathcal{K}_{\mathcal{T}(w)} \subset \mathcal{S}$ by contradiction. Suppose there exists a $\widehat{\theta}_j(w)$ such that $\widehat{\theta}_j(w) \in \mathcal{K}_{\mathcal{T}(w)}$ but $\widehat{\theta}_j(w) \notin \mathcal{S}$. Since $\widehat{\theta}_j(w) \notin \mathcal{S}$, then $\widehat{\theta}_j(w) \in \mathcal{T}(w) \setminus \mathcal{S}$. This means that $\forall \widehat{\theta}_i(w) \in \mathcal{S}$ we have $\widehat{\theta}_i(w) > \widehat{\theta}_j(w)$, and since $|\mathcal{S}| = k$, this implies that $\mathrm{rank}(\widehat{\theta}_j(w); \mathcal{T}(w)) > k$, contradicting the assumption $\widehat{\theta}_j(w) \in \mathcal{K}_{\mathcal{T}(w)}$. $\qquad \square$

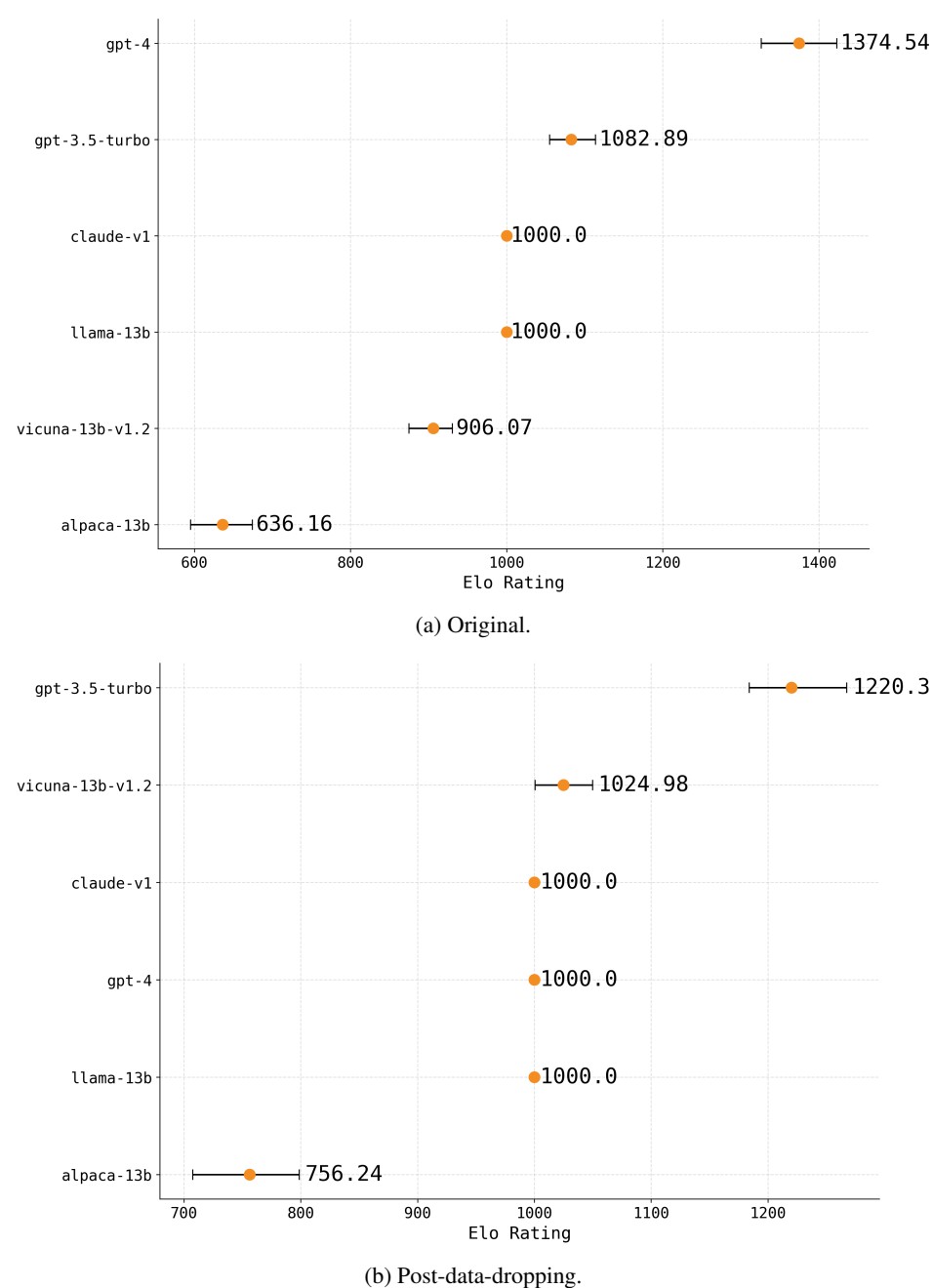

(a) Original.

(b) Post-data-dropping.

Figure 9: 95% Bootstrap-confidence-interval-based rankings on MTBench (LLM Judge).

## C    AMIP APPROXIMATION FOR BT MODELS

### C.1    AMIP APPROXIMATION OF GENERAL WEIGHTED BT MODELS

For completeness we provide here a review on general AMIP approximation proposed by Broderick et al. (2020) to solve the optimization problem Equation (8).

Broderick et al. (2020) propose relaxing $w$ to allow continuous values and replacing the $w$-specific quantity of interest with a first-order Taylor series expansion with respect to $w$ around $1_N$. This first-order Taylor series expansion is known as the *influence function (IF)* approximation (Hampel et al., 2011), a classic technique from robust statistics that approximates the affect of upweighting

| Arena | Fraction of Trials Top-1 Robust |
|---|---|
| Chatbot Arena (Human-judge) | 0.77 (0.97 at $\alpha = 0.1\%$) |
| Vision Arena | 1.00 |
| NBA Games | 1.00 |
| Chatbot Arena (LLM-judge) | 1.00 |
| Webdev Arena | 1.00 |
| Search Arena | 1.00 |
| MT-bench (LLM-judge) | 1.00 |
| ATP Tennis | 1.00 |
| MT-bench (Human-judge) | 1.00 |

Table 3: Top-1 robustness of each arena under uniform-at-random data-dropping. Each entry reports the proportion of 100 trials in which dropping $1\%$ of the evaluations uniformly-at-random does not change the top-ranked model.

(or dropping) a data point on model parameters using a first-order Taylor series approximation in data-weight space. Influence functions have become popular tools for approximating resampling methods (Giordano et al., 2019) and assigning value to data that a model was trained on (Koh & Liang, 2017; Park et al., 2023). This approximation applies to more general data analyses and quantities of interest.

In our case, this approximation amounts to replacing Equation (8) with

$$\max_{w \in W_\alpha} \sum_{n=1}^{N} (1 - w_n) \left( \left. \frac{\partial \hat{\theta}_i(w)}{\partial w_n} \right|_{w=1_N} - \left. \frac{\partial \hat{\theta}_j(w)}{\partial w_n} \right|_{w=1_N} \right). \tag{10}$$

Let

$$L(y_n, \theta) := w_{WL} I_{y_n=W} \log \sigma(\theta_{i_n} - \theta_{j_n}) + w_{WL} I_{y_n=L} \log(1 - \sigma(\theta_{i_n} - \theta_{j_n}))$$
$$+ w_T I_{y_n=T} \big( \log \sigma(\theta_{i_n} - \theta_{j_n}) + \log(1 - \sigma(\theta_{i_n} - \theta_{j_n})) \big). \tag{11}$$

to be the likelihood for a single data point. The impact of upweighting $w$ on the parameter $\hat{\theta}_i(w)$ is then given by

$$\left. \frac{\partial \hat{\theta}_i(w)}{\partial w_n} \right|_{w=1_N} = -H_{\hat{\theta}(1_N)}^{-1} \left. \nabla_\theta L(y_n, \theta) \right|_{\theta=\hat{\theta}(1_N)}, \tag{12}$$

where

$$H_{\hat{\theta}(1_N)} := \frac{1}{N} \sum_{n=1}^{N} \left. \nabla_\theta^2 L(y_n, \theta) \right|_{\theta=\hat{\theta}(1_N)}. \tag{13}$$

See Broderick et al. (2020, Section 2.2.2) for more details on this derivation. In what follows we provide details on how to apply this approximation in BT models by reformulating it as a logistic regression.

## C.2 BT MODELS AS LOGISTIC REGRESSIONS

**Unweighted BT.** In the unweighted BT model with $w_{W,L} = 1, w_T = 0$, with an abuse of data indices $n$, the preferences are assumed to be generated as

$$y_n \sim \text{Bernoulli}(\sigma(\theta_{i_n} - \theta_{j_n})), \tag{14}$$

We can cast this model as a logistic regression with a specially-structured design matrix. We denote the corresponding "design" vector of the $n$th comparison, $x_n \in \{-1, 0, 1\}^M$, a vector encoding which two players are being compared. That is, if the game is between players $i$ and $j$, then $x_n$ has a 1 in the $i$th element, a $-1$ in the $j$th element, and 0 otherwise. Using this structure, we can rewrite the model as a logistic regression model with $M - 1$ parameters corresponding to the scores of the players, $\boldsymbol{\theta} = (\theta_1, \ldots, \theta_M) \in \mathbb{R}^M$ with $\theta_1 = 0$,

$$y_n \sim \text{Bernoulli}(\sigma(x_n^\top \boldsymbol{\theta})). \tag{15}$$

We fit the BT-model (i.e., estimate $\boldsymbol{\theta}$) by maximum likelihood of logistic regression,

$$
\hat{\boldsymbol{\theta}} := \underset{\boldsymbol{\theta}:\theta_1=0}{\arg\max} \sum_{n=1}^{N} \big( y_n \log \sigma(x_n^\top \boldsymbol{\theta})
$$
$$
+ (1 - y_n) \log(1 - \sigma(x_n^\top \boldsymbol{\theta})) \big). \tag{16}
$$

**Weighted BT.** The model actually used in e.g., ChatBot Arena that handles tie by 1) counting every winning/loss as two games with the same outcome and 2) couting tie as two games with opposite outcomes. This effectively sets $w_{W,L} = 2, w_T = 1$. This special case can also be casted as a logistic regression with two copy of the design matrix same as unweighted version, $\boldsymbol{X}_{weighted} = [\boldsymbol{X}, \boldsymbol{X}]$. That is, suppose there are in total $N$ games, if the $n$th game is between players $i$ and $j$, then $x_{weighted,n}$ as well as $x_{weighted,n+N}$ has a 1 in the $i$th element, a $-1$ in the $j$th element, and 0 otherwise. The response $y_{weighted,n} = I_{y_n=W}$ and $y_{weighted,n+N} = I_{y_n=W} + I_{y_n=T}$. I.e., in the first copy of the game, a tie is counted as a loss and in the second copy of the game, a tile is counted as a win while winning and losing are counted twice in total from both copies. Then we can fit the weighted BT by maximum likelihood of logistic regression,

$$
\hat{\boldsymbol{\theta}} := \underset{\boldsymbol{\theta}:\theta_1=0}{\arg\max} \sum_{n=1}^{2N} \big( y_{weighted,n} \log \sigma(x_{weighted,n}^\top \boldsymbol{\theta})
$$
$$
+ (1 - y_{weighted,n}) \log(1 - \sigma(x_{weighted,n}^\top \boldsymbol{\theta})) \big). \tag{17}
$$

### C.3 Applying AMIP to BT models in logistic form

In this section we provide details on applying general Equation (12) in our specific case of logistic regression formed BT models. We observed that our quantity of interest $\theta_i - \theta_j$ is a linear combination of effect size $\theta_i$s in logistic regression, thus the first order Taylor expansion of this quantity can be calculated by first order Taylor expansion of $\theta_i$s.

Let $e_j$ denote the $j$th standard basis vector and $\mathbf{X} \in \mathbb{R}^{N \times P}$ denote the design matrix. Let $\widehat{p}_n = \sigma(\hat{\theta}^\top x_n)$ and $\boldsymbol{V} = \text{diag}(\{\widehat{p}_n(1 - \widehat{p}_n)\}_n)$. For logistic regression with an effect-size quantity of interest, $\theta_j$, the formula for the influence score for the $n$th data point (Pregibon, 1981) is given by

$$
\frac{\partial \hat{\theta}_j(w)}{\partial w_n}\bigg|_{w=1_N} = e_j^\top (\mathbf{X}^\top \boldsymbol{V} \mathbf{X})^{-1} x_n \widehat{p}_n(1 - \widehat{p}_n)(y_n - \widehat{p}_n), \tag{18}
$$

In addition to influence functions, our framework enables a second data-dropping approximation known as the *One-step Newton (1sN)* approximation, which approximates the effect of dropping a data point on model parameters using a second-order Taylor expansion in parameter space. This Newton-style update has become popular for approximating the deletion of data in recent works on approximate cross validation (Ghosh et al., 2020; Wilson et al., 2020) and machine unlearning (Sekhari et al., 2021; Suriyakumar & Wilson, 2022). The 1sN is slightly more expensive to compute than the IF approximation (as it corrects the IF with a multiplicative correction term) but is more accurate when the to-be-dropped data point has high a leverage score (because the correction term involves the leverage score of a data point). Previous works have proposed approximating the removal of a group of data points by the sum of leave-one-out 1sN scores, in an algorithm known as the **Additive one-step Newton approximation** (Huang et al., 2025a; Park et al., 2023).

To run the AMIP and Additive one-step Newton algorithm to check pairwise robustness between two given players, $i$ and $j$, we:

1. Fit a BT model on the entire arena.
2. Compute the *influence scores* (Equation (18)) (one-step Newton scores for the Additive one-step Newton algorithm) for all matches in the arena.
3. Identify the $\lfloor \alpha N \rfloor$ matchups for which the difference in influence scores is the largest in the negative direction (assuming that player $i$ has a higher estimated BT score than player $j$ on the full data).
4. Approximate impact of dropping these $\lfloor \alpha N \rfloor$ matchups by the sum of the influence score approximations.

5. If the approximation predicts that the relative ranking between players $i$ and $j$ changed, then refit the model leaving out the identified subgroup.[7]

These data-dropping algorithms replace a computationally intractable combinatorial search with an algorithm that costs only

$$O(Analysis + N \log(\alpha N) + NP^2 + P^3),$$

where $Analysis$ represents the cost of fitting the initial Bradley–Terry model on the original arena to compute scores. Data-dropping approximations make identifying candidate subsets of the arena that may induce top-$k$ non-robustness very fast because they eliminate the need to retrain the BT model repeatedly on every candidate subset. Once a candidate subset is identified, however, our method always performs a *refitting* of the BT model with the identified subset removed to verify whether the non-robustness is true. This final verification step ensures that our method does not return false positives.

## D  ARENAS

**Chatbot Arena.** A crowdsourced platform where users engage in conversations with two chatbots at the same time and rate their responses based on personal preferences (Zheng et al., 2023). We use the `arena-human-preference-55k` amd `chatbot-arena-llm-judges` datasets. This benchmark contains a total number of 57,477 preferences. Figure 3 presents the BT scores of the top models in Chatbot Arena.

**MT-Bench.** A multi-turn question set designed to compare LLMs in multi-turn conversation and instruction following constructed to distinguish between models based on reasoning and mathematics (Zheng et al., 2023). We use the `mt-bench-human-judgments` dataset. This benchmark was handcrafted using 58 expert-level human labelers; it contains 3,355 total preferences. In contrast to Chatbot Arena, labelers are mostly graduate students, so they are considered more skilled than average crowd workers. Figure 8 presents the BT scores of the models in MT-bench.

**Search Arena.** A crowdsourced platform for search-augmented LLMs, focusing on real-world and current events rather than static factual questions. We conduct our analysis using historical data available on Hugging Face: `lmarena-ai/search-arena-24k`. The dataset contains 24,069 multi-turn conversations with search-LLMs across diverse intents, languages, and topics. Figure 6 presents the BT scores of the top models in Search Arena.

**Webdev Arena.** A crowdsourced platform for LLM web development tasks, such as building interactive applications and webpages. We conduct our analysis using historical data available on Hugging Face: `lmarena-ai/webdev-arena-preference-10k`. This dataset contains 10,000 user-submitted prompts. Figure 7 presents the BT scores of the top models in Webdev Arena.

**Vision Arena.** A crowdsourced platform that tests vision-language models on visual question-answering. There are a total of 30,000 single and multi-turn chats between users and two anonymous vision-language models. We conduct our analysis using historical data available on Hugging Face: `lmarena-ai/VisionArena-Battle`. Figure 5 presents the BT scores of the top models in Vision Arena.

**ATP Tennis.** Association of Tennis Professionals (ATP) tennis records consolidated by Sackmann (2024). Each entry represents a match from the ATP tour, a worldwide top-tier men's tennis tour, and consists of the identifiers of the winning and losing players and the match-related metadata (e.g., player rankings, name of the tournament). We focused on the top-10 ranked players based on the 2024 season ranking and analyzed their plays throughout four seasons, 2020-2024. To avoid the case where dropping a small proportion of matches could drop a player's entire record, we focus our analysis on players who played at least 20 games. There were in total 278 games after filtering. Figure 10 presents the BT scores of the top models in the tennis dataset.

**NBA.** Basketball games from all seasons of the National Basketball Association (NBA), consolidated by FiveThirtyEight (2025). Each entry represents a historical game from the National Basketball Association, consisting of the identifiers of the two teams, the outcome of the game (win or

---

[7]Our algorithm gives users the option to refit the BT model for all matchups, regardless of whether a predicted ranking change occurs.

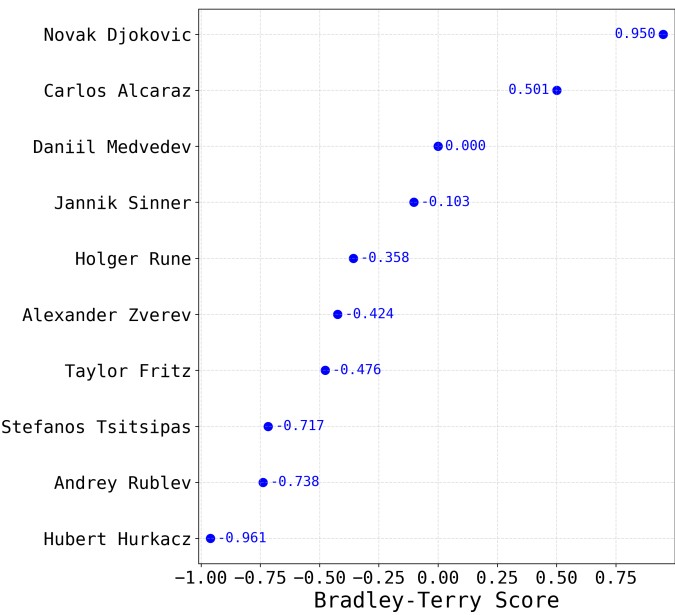

Figure 10: The top-10 player rankings in the tennis data.

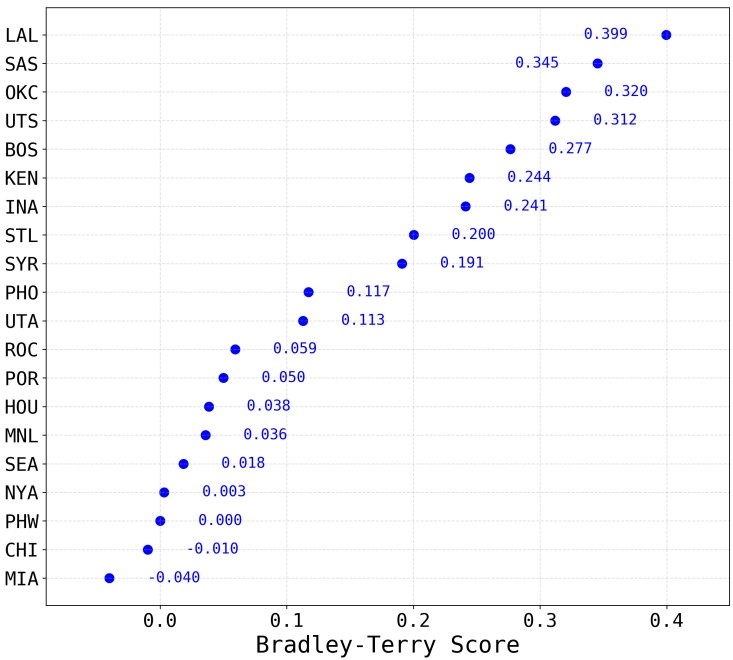

Figure 11: The top-20 team rankings in the NBA.

loss), as well as game-related metadata (e.g., Elo score of each team, game location). To avoid the case where dropping a small proportion of matches could drop a player's entire record, we focus our analysis on the top 50 teams by number of games played. There are a total of 109,892 matchups between the 50 teams. Figure 11 presents the BT scores of the top teams in the NBA.

# E   Player Involvement, Homogeneous Bars

Across all top-$k$ robustness experiments, $100\%$ of dropped matches involved either one or both of the models whose rankings were flipped, with $100\%$ belonging to one of these two cases within a given $k$ (see Figure 12). There are no partial bars or mixed compositions. Readers may ask: Why does this homogeneous pattern consistently appear? Could this be a property of the arena data?

We investigate this by manually inspecting the dropped matchups returned by our robustness assessing algorithm for each value of $k$. Specifically, in each case, we identified the dropped matchups and inspected which players appeared in these matchups. We summarize the findings here:

- $k = 1$: 2 games were dropped to flip GPT-4-0125-preview (originally 1st) and GPT-4-1106-preview (2nd). These two matches were between GPT-4-0125-preview and two other models, vicuna-13b (22nd) and stripedhyena-nous-7b (45th), with GPT-4-0125-preview losing.

- $k = 3$: 29 games were dropped to flip models gpt-4-0314 (3rd place) with mistral-7b-instruct-v0.2 (6th place). Games were played between mistral-7b-instruct-v0.2 and various other models, with mistral-7b-instruct-v0.2 losing all matches.

- $k = 5$: 3 games were dropped to flip models qwen1.5-72b-chat (5th place) with mistral-medium (6th place). All dropped matches were between qwen1.5-72b-chat and gpt-4-1106-preview (1st place), with qwen1.5-72b-chat (5th place) winning.

- $k = 10$: 1 game was dropped to flip models gemini-pro (10th) and mixtral-8x7b-instruct-v0.1 (11th place). The dropped match was between the two models, with gemini-pro winning.

- $k = 20$: 1 game was dropped to flip models gpt-3.5-turbo-0314 (20th place) with nous-hermes-2-mixtral-8x7b-dpo (21st place). The dropped match was between nous-hermes-2-mixtral-8x7b-dpo (21st place) and vicuna-13b (22st place), with nous-hermes-2-mixtral-8x7b-dpo losing.

The reason the involvement is always entirely either one or both affected players is because all of the dropped matchups consist of games played between a central model and a specific competitor (or group of competitors) whose outcomes all favor or disfavor the specific model and every dropped preference was a clear win or loss (no ties), aligning in the direction required to flip the ranking. In other words, whenever a top-$k$ set changed due to the demotion of a model, all dropped matches were ones that the demoted model had originally won, and vice versa for promotions. This structure then leads the dropped matchups to consist entirely of evaluations that involved one or both ranking-flipped models. This finding reveals something interesting about the nature of the non-robustness in our analysis: small, consistent sets of matchups are sufficient to push a model just above or below another on the leaderboard.

For every instance where the top-$k$ leaderboard changes due to dropped preferences, we find that the affected matches always involve at least one of the models whose rank is altered (see Figure 12). This holds true for both human-judged and LLM-judged Chatbot Arenas. While Min et al. (2025) find that adding in a small fraction of rigged votes can influence a target model's ranking even when the target model is not directly involved in the rigged votes, we are unable to find instances where rankings were flipped by removing a small fraction of preferences where neither of the affected models were involved.

Also, notice in Figure 12 that there are no partial bars or mixed compositions. We investigate why this homogeneous pattern appears consistently across bars. Inspecting dropped matchups manually, we find that the reason why one or both flipped players are always involved in the dropped matchups is because these matchups are always played between the model that is flipped, call it the target model, and a specific competitor (either the model whose ranking is flipped relative to the target model, or another model) or group of competitors (including models whose rankings remain unchanged), and all matchups either always favor or disfavor the target model (see Appendix E for a more detailed description). This finding reveals something about how non-robustness appears in our analyses: small, consistent sets of matchups are sufficient to push a model just above or below another on the leaderboard.

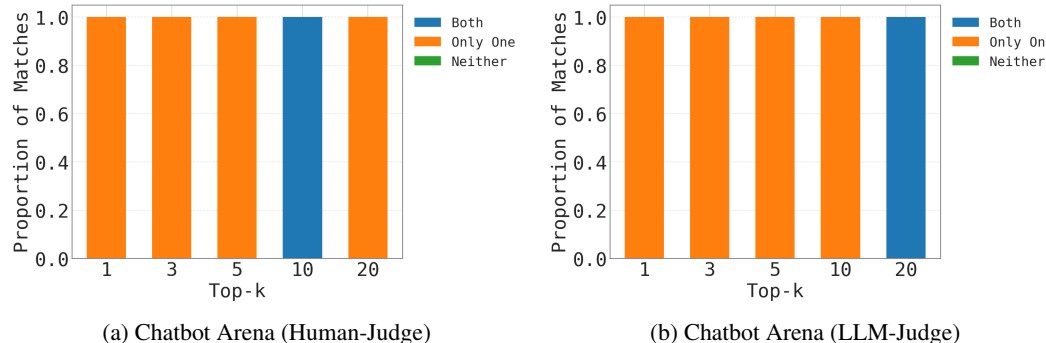

(a) Chatbot Arena (Human-Judge)    (b) Chatbot Arena (LLM-Judge)

Figure 12: *Player involvement* in the most influential matches whose removal caused two models (players), one inside the top-$k$ positions and one outside, to exchange places. Each bar represents the composition of matches dropped in order to exchange the models. The proportions indicate whether neither (green), one (orange), or both (blue) of the affected models were involved in each dropped matchup. For Chatbot Arena (Human-Judge) (top), the number of matches dropped for each $k \in \{1, 3, 5, 10, 20\}$ is $(2, 29, 3, 1, 1)$, respectively. For Chatbot Arena (LLM-Judge) (bottom), the number of matches dropped for each $k \in \{1, 3, 5, 10, 20\}$ is $(9, 24, 9, 1, 2)$. Across all $k$, we find that $100\%$ of flipped rankings involved matches containing at least one of the players whose ranking changed.

## F    INSPECTING DROPPED PREFERENCES

Dropping just two ($0.003\%$ of) preferences on Chatbot Arena is enough to change the top-ranked model on Chatbot Arena from GPT-4-0125-preview to GPT-4-1106-preview. Below we provide the two prompts and response pairs responsible for the ranking flip, together with the corresponding annotation. Additionally, we note that the BT model's estimation procedure does not use any information about prompt content; it depends only on the BT scores of the two models involved in each match, which means it is possible for dropped subsets to be non-unique.

**Prompt 1.**   "teach me how to make independent classes in python that can be call and install through pip."

**Competing Models.**   GPT-4-0125-preview, Vicuna-13b

**Preferred Model.**   *Vicuna-13b.*

**Responses.**   See Figure 13 for the complete response by GPT-4-0125-preview and Figure 14 for the complete response by Vicuna-13b.

**Prompt 2.**   "name me challenging c++ projects i can add on my cv as a cs student. give me creative ideas and dont forget im a student."

**Competing Models.**   GPT-4-0125-preview, Stripedhyena-nous-7b

**Preferred Model.**   *Stripedhyena-nous-7b.*

**Responses.**   See Figure 15 for the complete response by GPT-4-0125-preview and Figure 16 for the complete response by Stripedhyena-nous-7b.

To conduct a qualitative analysis on the response pairs, we provide a strong judge model, GPT-5.1[8], with a system prompt requesting a summary of the two responses, their similarities and differences, and a difficulty rating indicating how distinct the responses are.

---

[8]We accessed GPT-5.1 through the ChatGPT user interface in a logged-out, incognito session.

**System Prompt**

Provide a summary of the following two AI-generated responses to the prompt: [Insert Prompt Here]. Then describe their main similarities and differences. Finally, rate on a scale from 1 (very similar) to 5 (one response clearly better) how difficult it would be for a user to distinguish the two responses.

**Response A:** [Insert Response A here]
**Response B:** [Insert Response B here]

Provide 2-3 sentences for each of the following headings: summary, similarities, differences, and difficulty rating.

In the following GPT-5.1 responses, Response A refers to GPT-4-0125-preview, and Response B refers to the smaller, open-source models, Vicuna-13b and Stripedhyena-nous-7b, respectively for prompts 1 and 2.

**GPT-5.1-Generated Response [for Prompt 1]**

**Summary.** Response A provides a detailed, step-by-step explanation of how to create Python classes, organize them into a package, build distribution files, and optionally upload the package to PyPI. It includes full code examples, terminal commands, and practical guidance for beginners. Response B offers a brief overview of how to create a package with init.py and setup.py, focusing mainly on local installation and basic configuration, and ends abruptly without deeper detail.

**Similarities.** Both responses explain that Python packages require an init.py file and a setup.py file using setuptools. They also describe how a user can install the finished package with pip. Each provides an example of what a setup.py file might look like.

**Differences.** Response A is much more comprehensive, guiding the reader through class creation, directory structure, building distribution files, and publishing to PyPI, while Response B offers only a high-level outline of the process. Response A includes practical, runnable examples, whereas Response B only sketches the idea of the setup. Response B also feels incomplete and omits several important steps that Response A covers thoroughly.

**Difficulty Rating.** On a scale of 1 to 5, where 1 means the responses are very similar and 5 means one is clearly better, the two responses would rate around a 4.5. A typical user would likely find Response A significantly more helpful, complete, and actionable than Response B.

**GPT-5.1-Generated Response [for Prompt 2]**

**Summary.** Response A focuses on C++-specific, systems-oriented projects such as memory allocators, compilers, and physics engines, emphasizing core CS concepts and low-level programming. Response B centers on AI- and application-focused ideas like chatbots, image recognition, and recommendation systems, many of which are not inherently tied to C++.

**Similarities.** Both responses offer lists of challenging project ideas intended to help a CS student strengthen their CV. They also highlight projects that can demonstrate technical skills and encourage documenting the work effectively.

**Differences.**    Response A is tightly aligned with the prompt, emphasizing C++ and foundational CS mechanics, whereas Response B presents more general software and AI projects that may rely on other languages or frameworks. Additionally, Response A provides deeper technical context, while Response B takes a more high-level, application-oriented approach.

**Difficulty Rating.**    Rating: 5 — The two responses differ clearly in focus, depth, and C++ relevance, making them easy to distinguish. A typical user would quickly notice that only Response A directly addresses the C++-project requirement.

In both surfaced examples, GPT-5.1 judges the pair of responses to be easy to differentiate and consistently prefers the opposite response from the human annotator (e.g., "A typical user would likely find Response A significantly more helpful, complete, and actionable than Response B," and "A typical user would quickly notice that only Response A directly addresses the C++-project requirement."). This makes sense, as both matches are cases in which a much lower-scoring model is preferred to the top-ranked model. Thus, one might interpret the influential subsets the method identifies as "outlier" preferences, cases where the annotator's preference deviates from what the average user might select.

## F.1    Sensitivity Driven by Narrow Score Margins

We find that the stability of the arena depends on the BT score margins between models (see Figure 18). Recall from Table 1 that dropping only two preferences is enough to change the top-ranked model. To explore the effect of score margins, we first remove all games involving the second-place model (GPT-4-1106-preview). The arena then becomes more resilient, requiring dropping 38 out of $57{,}477$ ($0.07\%$) preferences to overturn the leader. When we further remove all games involving the 2nd through 5th place models, the leaderboard becomes harder to perturb, but is still remarkably sensitive, requiring dropping 63 out of $57{,}477$ ($0.1\%$) preferences to flip the top model.

One possible explanation for this fragility is that top competitors are often closely matched, making it difficult for annotators to reliably separate their performance on the prompts submitted to the arena. This raises the possibility that sensitivity could be reduced by that sharpens distinctions between models (for example, through expert annotators and curated prompts targeting challenging domains such as mathematics, coding, and multi-turn reasoning, as in MT-Bench (Zheng et al., 2023)).

## G    Verification of AMIP-identified Subsets

For each of our nine data analyses, we can use the machinery of our method to return a $w$ corresponding to a smallest data subset that can be dropped to change the top-1 ranking. The machinery of our method also returns an estimate (before re-running the BT model) for whether the top-1 ranking is changed. To examine how often an identified weight vector $w$ truly corresponds to a subset whose removal flips the ranking, we report across all nine arenas the number of cases where the estimate with the $w$-vector accurately reflects a change in the top ranking over the total number of arenas tested for top-1 robustness.

In Table 4, we find that all identified $w$-vectors lead to a true change in ranking. And we find that this result holds even when the dropped subset is greater than $\lfloor \alpha N \rfloor$ of the data (even though the original AMIP makes no claims to an accurate identification of a decision-changing $w$ in this case). However, this does not mean that there are no cases where AMIP fails to surface a vector $w$ that leads to a change in ranking (i.e., false negatives are possible).

## H    Masking Effects and the Possibility of False Negatives

A main limitation of our approach is that while it can conclusively identify non-robustness, it is possible that there is non-robustness that it does not find: when our method surfaces a subset whose removal flips the ranking, the resulting perturbation is an exact, verifiable witness of fragility; however, when no such subset is found, we cannot conclude that the arena is robust.

| Dataset | AMIP-Returned Subset (Indices) | Flip? |
|---|---|---|
| Chatbot Arena | {46592, 5156} | Yes |
| Vision Arena | {22176, 9686, 887, 15782, 24340, 25110, 9816, 10926, 18732, 21303, 13957, 2934, 2936, 19600, 11072, 15311, 11038, 25845, 17732, 29100, 5421, 24462, 23006, 10572, 2134, 13518, 5390, 15353} | Yes |
| NBA Games | {18819, 19717, 18818, 19762, 14523, 19763, 14522, 20900, 22132, 22133, 18305, 15756, 14383, 18304, 14382, 19716, 20135} | Yes |
| Chatbot Arena (LLM-judge) | {41445, 9108, 14834, 11144, 11675, 9123, 17291, 48894, 42411} | Yes |
| Webdev Arena | {7164, 7539, 9112, 7711, 2089, 1815, 2414, 6542, 6446, 4883, 8753, 2889, 9272, 3553, 1512, 5933, 6992, 10387} | Yes |
| Search Arena | {22164, 12847, 12819, 21810, 11852, 19956, 9492, 15447, 11324, 16583, 12733, 10116, 21940, 15552, 9451, 12602, 21977, 11499, 12576, 10146, 12557, 11519, 15699, 9420, 12851, 18068, 12931, 11278, 13279, 11143, 11163, 21587, 9963, 13226, 9586, 20632, 13191, 9978, 13189, 12456, 11204, 17160, 13129, 18238, 18231, 10009, 13112, 15234, 11251, 20575, 13043, 10030, 11209, 9607, 20336, 15733, 22646, 12061, 11768, 12023, 10375} | Yes |
| MT-bench (LLM-judge) | {646, 587, 1290, 1741, 720, 570, 571, 72, 223, 1212, 1183, 1122, 2052, 2053, 2112, 1242, 1063, 1033, 1032, 1003, 1812, 2113, 1002, 282, 1093, 1092, 1243, 2022, 1753, 1752, 132, 103, 102, 1872, 1873, 1543, 162, 1453, 1423, 1422} | Yes |
| ATP Tennis | {236, 168, 251, 177, 202, 122} | Yes |
| MT-bench (Human-judge) | {137, 2399, 1298, 1884, 2398, 139, 1153, 850, 391, 1111, 3181, 91, 648, 2612, 803, 802, 804, 801, 800, 348, 744, 41, 2726, 349, 2668, 608, 607, 1450, 799, 2909, 1409, 2912, 2725, 748, 2492, 1537, 160, 1536, 2911, 1534, 925, 1535, 2333, 2161, 570, 1830, 346, 2334, 745, 1408, 1191, 2332, 3055, 101, 222, 2883, 3274, 221, 2837, 219, 667, 178, 3021, 3022, 1902, 2552, 2551, 2341, 863, 1124, 1903, 2624, 2626, 2627, 1634, 898, 1744, 2510, 1745, 220, 3275, 666, 1162, 246, 1214, 1294, 1165, 64, 247, 1556, 65, 3278} | Yes |

Table 4: For each dataset, the number of cases where the estimate with the $w$-vector accurately reflects a change in the top ranking) over the total number of arenas tested for top-1 robustness. All surfaced $w$-vectors successfully flip the ranking (9/9).

This limitation is a documented challenge in the literature on identifying influential subsets (Hu et al., 2024; Huang et al., 2025a; Moitra & Rohatgi, 2023). In linear models, for example, Huang et al. (2025a) and Hu et al. (2024) show that AMIP and related additive, first-order approximations can miss influential subsets. A key failure mode is due to a phenomenon is known as "masking," in which several highly-impactful data points produce a large change to a statistic (e.g., an estimated BT-score) when deleted jointly, yet no single point appears influential when considered in isolation. To address masking effects, works such as Belsley et al. (1980); Kuschnig et al. (2021); Huang et al. (2025a) have considered using step-wise (greedy) approaches, of removing the most influential points in sequence. The main empirical conclusion of this paper relies on an existence proof: namely, that several widely used evaluation datasets exhibit substantial sensitivity to very small targeted deletions. For this reason, we do not pursue step-wise greedy variants here, though they remain an interesting direction for future work.

## I NON-ROBUSTNESS OF NBA RANKINGS

To assess whether fragility of Bradley–Terry (BT) rankings extends beyond LLM arenas, we applied our method to historical NBA data. The degree of non-robustness in NBA rankings is comparable to that observed in Chatbot Arena: both require removing $< 0.05\%$ of matchups to alter the top spot. In both cases, the explanation may trace back to small BT-score gaps at the top of the leaderboard (See Figure 11). One plausible explanation is that the skill levels among elite NBA teams are tightly clustered, and so any apparent differences in skill level may not be substantive.

This non-robustness in the NBA datasets suggests a broader conclusion that BT-based evaluation systems–whether in AI benchmarking or sports–tend to be unstable when the margin between competitors is narrow.

## J    ADDITIONAL SUPPORTING FIGURES

The figures in this section provide additional insights related to our analysis. Figure 17 shows the distribution of model appearances in Chatbot Arena, respectively, revealing differences in evaluation density and coverage across platforms. Figure 18 illustrates the relationship between the robustness of model rankings and the BT score gap between adjacent models, confirming that small score differences tend to coincide with greater sensitivity to worst-case data-dropping.

## K    LARGE LANGUAGE MODEL (LLM) USE

We used LLMs for grammar checks and to polish writing, to help find sports datasets that yielded the discovery of Sackmann (2024), and as a judge model in the qualitative study described in Appendix F. Although our study is about LLM rankings, we did not use LLMs as direct study subjects.

"Creating independent classes in Python that can be called and installed
through `pip` involves a few steps. You'll be packaging your code as a Python
package, and optionally, distributing it through PyPi (Python Package Index)
so that others can easily install it using `pip`. Let's go through the
process.### Step 1: Organize Your Code1. **Create a Project Directory**: This
is where your package code will reside. Choose a unique name for your
directory, preferably related to your package name.     ```bash   mkdir
mypackage   cd mypackage   ```2. **Create a Package Directory**: Inside your
project directory, create another directory with the name of your package.
This is where the Python files containing your classes will go.   ```bash
mkdir mypackage   ```3. **Write Your Classes**: Inside the package directory,
create Python files (`*.py`) where you'll define your classes. For example,
create `classA.py` and define a class `ClassA` in it.   ```python   #
mypackage\/mypackage\/classA.py      class ClassA:      def __init__(self):
print(\"Class A is initialized\")               def greet(self):
return \"Hello from Class A\"   ```### Step 2: Make Your Package Installable1.
**Create a `setup.py` File**: At the root of your project directory (not
inside your package directory), create a `setup.py` file. This file includes
metadata about your package and the instructions to install it.   ```python
from setuptools import setup, find_packages   setup(      name=\"mypackage\",
version=\"0.1\",       packages=find_packages(),      description=\"A simple
example package\",      author=\"Your Name\",
author_email=\"your.email@example.com\",
url=\"https:\/\/github.com\/yourusername\/mypackage\",      classifiers=[
\"Programming Language :: Python :: 3\",          \"License :: OSI Approved
:: MIT License\",          \"Operating System :: OS Independent\",      ],
)   ```2. **Create an `__init__.py` File**: Inside your package directory,
create an `__init__.py` file. This file can be empty but it is necessary to
make Python treat the directories as containing packages.   ```bash   touch
mypackage\/__init__.py   ```### Step 3: Build Your Package1. **Ensure
`setuptools` and `wheel` are Installed**: These packages are needed to build
your package.   ```bash   python3 -m pip install --user --upgrade setuptools
wheel   ```2. **Build Your Package**: From the root of your project directory,
run the following command to build your package.   ```bash   python3 setup.py
sdist bdist_wheel   ```This will generate distribution files in the `dist`
directory.### Step 4: (Optional) Publish Your Package to PyPI1. **Register an
Account on PyPI**: To distribute your package, you'll need an account on
[PyPI](https:\/\/pypi.org\/).2. **Install Twine**: Twine is a utility for
publishing Python packages on PyPI.   ```bash   python3 -m pip install --user
--upgrade twine   ```3. **Upload Your Package**: Use Twine to upload your
package to PyPI.   ```bash   python3 -m twine upload dist\/*   ```4. **Install
Your Package**: Once uploaded, anyone (including you) can install your package
using `pip`.   ```bash   pip install mypackage   ```### Step 5: Using Your
PackageAfter installing your package via `pip`, you can use it in your Python
scripts like so:```pythonfrom mypackage.classA import ClassAa =
ClassA()print(a.greet())```This guide provides a basic overview. Depending on
your needs, you might want to explore more advanced packaging options, such as
including dependencies, providing executable scripts, or supporting C
extensions."

Figure 13: GPT-4-0125-preview: response to prompt 1.

"To create a reusable Python package that can be installed through pip, you can follow these steps: 1. Create a new directory for your package.2. Inside the directory, create a file named `__init__.py` (note that the `__init__` part is important, as it tells Python that this is a package and not just a directory).3. In the `__init__.py` file, import any other modules that your package needs to function.4. Create a `setup.py` file in the same directory as your `__init__.py` file. This file will contain information about your package that will be used when it is installed through pip.5. In the `setup.py` file, specify the name of your package, a description of what it does, and any dependencies it has. You can use the `setuptools` library to help with this. For example:```from setuptools import setup, find_packagessetup( name='my_package', version='0.1', description='This is a description of my package', packages=find_packages(), install_requires=['numpy', 'pandas'],)```This tells pip that your package is called `my_package`, that it has a version number of `0.1`, and that it has two dependencies: `numpy` and `pandas`.6. Save the `setup.py` file and run the following command in your terminal to install your package:```pip install .```This will install your package and its dependencies in the current Python environment. You can then import and use your package in other Python projects.Note that this is just a basic example of how to create a Python package that can be installed through pip. There are many other options and considerations that you may want to take into account when creating a package, such as creating an `RE"

Figure 14: Vicuna-13b: response to prompt 1.

"Certainly! As a computer science student, engaging in challenging C++
projects can significantly enhance your CV by showcasing your coding skills,
problem-solving abilities, and creativity. Here are some creative and
challenging project ideas that span various levels of complexity:\n\n1.
**Custom Memory Allocator**: Develop a C++ project that involves creating a
custom memory allocator. This will help you understand how dynamic memory
allocation works under the hood and improve your knowledge of pointers, memory
management, and optimization techniques.\n\n2. **Blockchain Simulation**:
Implement a simple blockchain in C++. This project will introduce you to the
fundamentals of blockchain technology, including creating blocks, implementing
a proof-of-work algorithm, and understanding the security principles behind
blockchain.\n\n3. **Graphical User Interface (GUI) Chat Application**: Use C++
with Qt or any other GUI framework to create a desktop-based chat application.
This project can involve client-server architecture, understanding sockets,
and multi-threading to handle multiple clients.\n\n4. **Physics Simulation
Engine**: Build a physics engine that can simulate basic physics principles
like gravity, collision detection, and motion. This project can be extended by
incorporating a graphical representation of the simulation using libraries
such as OpenGL.\n\n5. **Machine Learning Algorithm Implementation**: Implement
a machine learning algorithm from scratch in C++. You could choose algorithms
like linear regression, decision trees, or even a simple neural network. This
will deepen your understanding of machine learning concepts and mathematical
principles.\n\n6. **Game Development with Unreal Engine**: Unreal Engine uses
C++ and offers an excellent platform to develop your own game. You could start
with something simple like a puzzle or platformer game and gradually add
complexity. This project will enhance your understanding of object-oriented
programming, game physics, AI, and real-time rendering.\n\n7. **Embedded
Systems Simulator**: Create a simulator for embedded systems that can model
the behavior of microcontrollers and peripherals. This project will introduce
you to the world of embedded systems, requiring you to understand low-level
programming and hardware-software interaction.\n\n8. **Compiler for a Simple
Language**: Design and implement a compiler for a simplified programming
language you create. This involves understanding lexical analysis, parsing,
syntax tree generation, and code generation. It's a challenging project that
will significantly improve your programming skills and understanding of how
compilers work.\n\n9. **File System Implementation**: Develop your own
simplified version of a file system in C++, which involves creating a way to
store, organize, retrieve, and manage data on a storage medium. This project
requires an understanding of disk management, directory structure, and file
metadata.\n\n10. **Augmented Reality App**: Utilize C++ with libraries such as
OpenCV and others to create a basic augmented reality application. This could
involve facial recognition, object tracking, or interactive games that overlay
digital content onto the real world.\n\nWhen selecting a project, consider
your interests and the specific skills you want to develop or showcase.
Completing one or more of these projects can make your CV stand out by
demonstrating your initiative, technical skills, and ability to tackle complex
problems."

Figure 15: GPT-4-0125-preview: response to prompt 2.

"1. AI-Powered Chatbot: Develop a chatbot that uses natural language processing and machine learning algorithms to simulate intelligent conversations with users. You can integrate it with popular messaging platforms like Facebook Messenger, Slack, or Telegram.\n\n2. Virtual Assistant: Create a virtual assistant application that can perform tasks like scheduling appointments, sending reminders, and providing recommendations based on user preferences and past behaviors.\n\n3. Image Recognition: Implement an image recognition system that can identify objects, people, and scenes from images. You can train the AI model using machine learning algorithms such as Convolutional Neural Networks (CNNs).\n\n4. Text-to-Speech Converter: Develop a text-to-speech converter that can convert written text into spoken words. This project will require knowledge of speech synthesis algorithms and text processing techniques.\n\n5. Recommendation System: Create a recommendation system for online shopping websites or streaming services. The system will analyze user preferences and behavior to suggest personalized products or content based on their interests.\n\n6. Augmented Reality (AR) Application: Design and develop an AR application that can enhance a user's real-world environment with virtual objects or information. This can be done using ARKit for iOS or ARCore for Android devices.\n\n7. Fraud Detection System: Implement a fraud detection system for e-commerce websites or banking applications. The system should be able to identify suspicious transactions based on user behavior and transaction patterns.\n\n8. Voice Recognition: Develop a voice recognition system that can accurately transcribe spoken words into text. This project will require knowledge of speech recognition algorithms and signal processing techniques.\n\n9. Game AI: Create an AI-powered game that adapts to the player's skills and preferences. You can use techniques like reinforcement learning to train the AI model to make intelligent decisions during gameplay.\n\n10. Chatbot Localization: Extend your chatbot project by adding support for multiple languages. This will require knowledge of natural language processing and translation algorithms.\n\nRemember to document your projects, highlight the technologies and programming languages you've used, and explain the challenges you've faced and how you've overcome them. Best of luck!"

Figure 16: Stripedhyena-nous-7b: response to prompt 2.

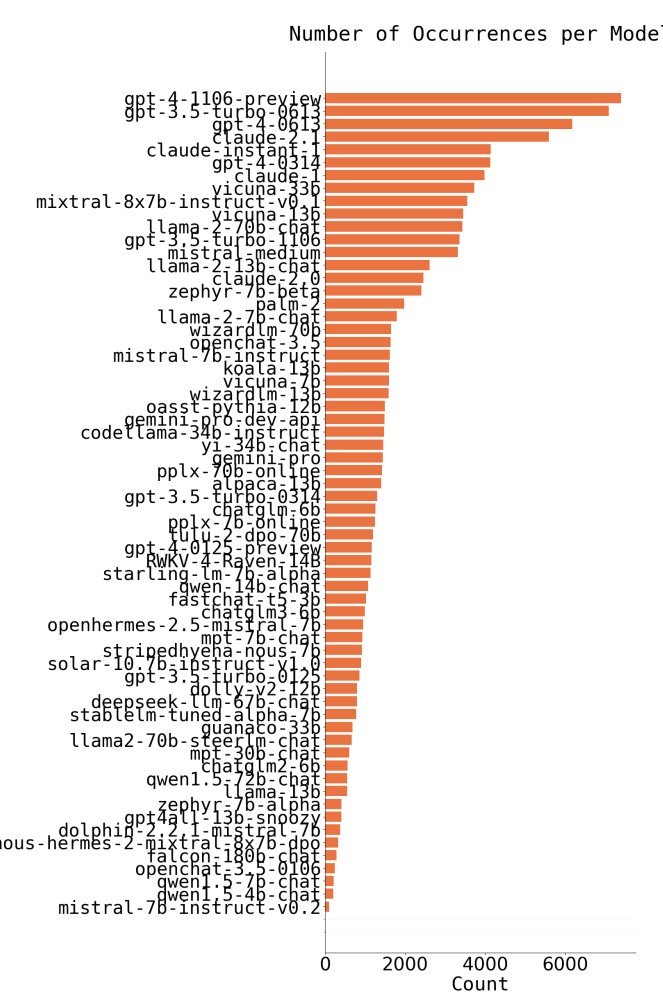

Figure 17: The number of times each model appears in a match in Chatbot Arena. The horizontal bar chart shows how frequently each model appeared in any match, with GPT-4 and GPT-3.5 variants being the most represented.

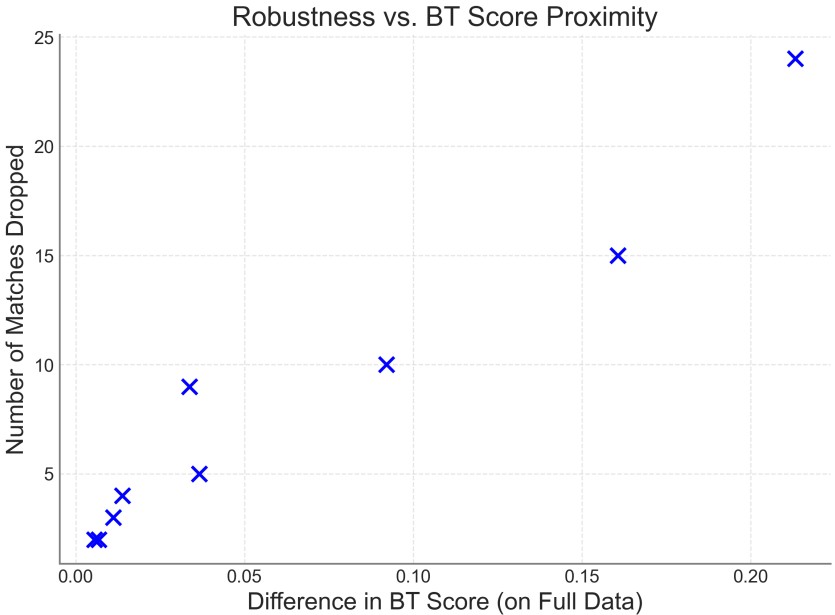

Figure 18: Robustness of results is correlated with the proximity of the BT scores. Each point represents a pair of models whose relative rankings flipped after dropping a small fraction of matchups. In every case, the flip causes one model to enter the top-k rankings (for some $k \in \{1, 3, 5, 10, 20\}$) while the other is demoted. These points are taken from both human and LLM-as-a-judge evaluation platforms.

