# OpenReview forum: "Dropping Just a Handful of Preferences Can Change Top Large Language Model Rankings"
_ICLR.cc/2026/Conference — ICLR 2026 Poster_

### Official Review · Reviewer_bN5G · 2025-10-30

**Soundness:** 4
**Presentation:** 4
**Contribution:** 4
**Rating:** 8
**Confidence:** 4

**Summary:**

This paper investigates the statistical robustness of LLM ranking systems based on the Bradley-Terry (BT) model, such as Chatbot Arena . The central research question is whether the top rankings on these leaderboards are stable or if they can be altered by removing a very small, worst-case fraction of the preference data. The authors propose a computationally efficient method, adapted from the Approximate Maximum Influence Perturbation (AMIP) literature, to identify the most influential preferences that, when dropped, are most likely to cause a ranking flip.

The authors show that popular LLM leaderboards are surprisingly fragile. Most strikingly, removing just two preference pairs, which account for only 0.003% of the data, was enough to change which model ranked first in Chatbot Arena. The analysis also shows that MT-Bench, which relies on expert reviewers and carefully chosen prompts, is much more stable than large crowdsourced platforms. Overall, the paper warns that rankings on many leaderboards may be less reliable than they appear, and that small score differences shouldn’t be overinterpreted.

**Strengths:**

**Originality**: The originality of this work lies in its novel research question and its surprising findings, rather than in the invention of a new statistical method. As far as I know, it is the first paper to systematically apply the concept of worst-case data-dropping robustness to the domain of major LLM leaderboards.

**Quality**: The paper shows high quality through its careful and valid application of statistical methods to real-world data. They verify the ranking flip by refitting the BT model. The use of public, real-world data enhances the work's credibility.

**Clarity**: The paper’s main finding is clearly reflected by its title. Figure 1 immediately conveys the core message to the readers. The results in Table 1 are clear and easy to interpret, which clearly supports the main point.

**Significance**: This work is a critical piece of scientific auditing for the entire LLM community. The finding that top rankings can be fragile has the potential to change community behavior, encouraging more skepticism and a demand for more robust evaluation practices.

**Weaknesses:**

Limited depth of explanation: The paper hypothesizes why MT-Bench is more robust (expert annotators, curated prompts) and why rankings are fragile (small BT-score margins), but these remain hypotheses. A more controlled experiment to disentangle these factors would be needed for a definitive causal claim.

Lack of methodological novelty: The weakness is that the core algorithm (AMIP) is adapted from prior work in statistics. The contribution is in the application and the discovery, not the invention of a new technique.

No solution to the problem: The paper is only a diagnostic tool, not a solution to the fragility of crowdsourcing rankings.

No average-case behavior: The paper focuses only on the worst case, not the average case (e.g., dropping random pairs of preferences), which I think is closer to the actual threat to these rankings, especially for crowdsourcing rankings.

**Questions:**

1. You qualitatively analyze the two prompts that flip the top ranking on Chatbot Arena and note that they involve "anomalous losses" against much lower-ranked models. Do you have a hypothesis about the nature of these prompts? Are they particularly tricky, ambiguous, or perhaps simple enough that the distinction between a top model and a lower-ranked model's response is negligible, leading to noisy human preferences?

2. Your findings suggest that small margins in Bradley-Terry scores are associated with reduced robustness. Could the confidence intervals of the BT scores themselves be used as a direct, and perhaps even simpler, proxy for ranking stability without needing to run your AMIP-based analysis?

3. Given your findings, what practical recommendations would you give to platforms like Chatbot Arena? Should they regularly run robustness checks like yours, or change the way ties are handled in the BT model?

4. When looking into the data, I also noticed that platforms like Chatbot Arena have highly unbalanced numbers of competitions across different model pairs (e.g., some model pairs have thousands of preference pairs, while many others have few or no pairs). Do you think this can explain their fragility? One extreme case could be removing a preference that favors the weak model over the strong model, even though the preference is the only one between the two models.

---

> ### Author Response · Authors · 2025-11-22
> **Response to reviewer bN5G (part I)**
>
> We thank the reviewer for pointing out the significance of this work as a critical piece of scientific auditing for the LLM community. Below, we provide responses to the reviewer’s questions and comments and refer the reviewer to the general reply for a high-level overview of the new additions.
>
> **(1) No average-case behavior.**
>
> To examine the contrast between worst-case data-dropping with dropping random pairs of preferences, we conduct a new uniform subsampling experiment (see Appendix A.2 of the manuscript). For each arena, we drop 1% of the evaluations uniformly at random, repeat the experiment 100 times, and record the fraction of runs in which the top-ranked model remains unchanged relative to the full arena. We present the results of this experiment in Table 3 in Appendix A.3. Across nearly all arenas, dropping 1% of the evaluations uniformly at random leaves the top-ranked model unchanged in every trial. Even Chatbot Arena (human-judge), which is the least stable under uniform subsampling, maintains its top-ranked model in 77% of random 1% deletions, a fraction that is many magnitudes larger than the 0.00348% of preferences required to flip the top-ranked model when dropping the worst-case data subset. These results show that the rankings are extremely sensitive to dropping a worst-case small fraction of preferences, yet stable (at $\alpha=1$%) to dropping preferences chosen at random. Taken together, these observations show that uniform and worst-case data-dropping probe fundamentally distinct failure modes.
>
> The reviewer raises a concern about the average case behavior as being the "actual threat to these rankings." We believe that this concern relies implicitly on the assumption that the sample of prompt-preference pairs on Chatbot Arena is representative of the underlying population of prompt-preference pairs that people ask chatbots in the wild. We think that this is a strong assumption to make for real-world preference datasets like Chatbot Arena, however, due to differences in annotators (e.g., the same, or similar types of, annotators may annotate several prompts on LMArena), resulting prompt-selection biases, and various other factors. For this reason, we might be worried if a very small fraction of prompt-preference pairs on the arena drive the top-rankings. To put it another way, we would be worried if we could drop a small, worst-case fraction of preferences on the arena and see resulting top rankings change, because what if Chatbot Arena contains (a very small fraction of) annotations that are not representative of what a typical user might prefer? In fact, in our analysis of dropped preferences in Appendix F, we do find instances where the influential surfaced preferences are ones that a strong judge model (GPT-5.1) deems to be atypical (different to what a "typical" user might prefer).

---

> > ### Author Response · Authors · 2025-11-22
> > **Response to reviewer bN5G (part II)**
> >
> > **(2) Hypothesis about the nature of prompts.**
> >
> > We thank the reviewer for this feedback. The two dropped prompts on Chatbot Arena are both related to programming, but they differ in intent: one asks a technical question about implementing Python classes, while the other requests creative project ideas in C++. Though both prompts fall under the general category of programming, we can not draw conclusions about the underlying cause of non-robustness. We have added a qualitative analysis on the identified response pairs in Appendix D and a reference to this analysis in Section 4.4. In this setup, we pass the response pairs to a strong judge model, GPT-5.1, with a system prompt requesting a summary of the two responses, their similarities and differences, and a difficulty rating indicating how distinct the responses are. In both surfaced examples, GPT-5.1 judges the pair of responses to be easy to differentiate and consistently prefers the opposite response from the original annotator. Thus, one might interpret the dropped preferences as being "outlier" judgments, cases where the annotator’s preference deviates from what the average user might select.
> >
> > We believe, however, that diagnosing the root cause of non-robustness lies beyond the scope of this work. Additionally, we note that the BT model’s estimation procedure does not use any information about prompt content; it depends only on the BT scores of the two models involved in each match, which means it is possible for dropped subsets to be non-unique. We have added this additional note in Appendix D (blue text). For this reason, we find it more meaningful to describe identified influential comparisons through the lens of the score differences among models involved in the dropped comparisons. Across all 36 matches surfaced while checking top-k robustness for on Chatbot Arena, every dropped preference was a clear win or loss (no ties), and every outcome aligned with the direction required to flip the ranking. We have added this discussion into Appendix C (see blue text).
> >
> > Our aim is to provide a tool that enables the community to create more principled follow-up studies to investigate the characteristics of influential prompts.
> >
> > **(3) Confidence Intervals.**
> >
> > We thank the reviewer for this feedback. To directly address the reviewer’s concerns, we added an experiment (see section A.1) where we applied our sensitivity analysis to the bootstrap-based rankings reported by LMArena [1]. Specifically, we computed the original bootstrap-based rankings (defined both in Chiang et al. and now in our paper’s Appendix A.1),  removed the influential comparisons identified by our method, and then recomputed the bootstrap-based rankings. The results of this new set of experiments are summarized in Table 2 in Appendix A.1. Even under the bootstrap-based rankings, we continue to find many arenas to be surprisingly sensitive to worst-case small-fraction data-dropping: the ranking flip persists even when the entire analysis is performed within the bootstrap framework. For instance, we surface arenas where the bootstrap-based ranking does output a single top-ranked model, but upon small-fraction data dropping, the model becomes no longer the sole top-ranked model (see Figure 7 in Appendix A.1). We include a discussion of this result and a reference to the experiment on bootstrap-based rankings in the main text Section 4.2. This shows that the non-robustness identified by AMIP is not an artifact of ignoring statistical uncertainty already captured by confidence intervals; the ranking flip persists even when the entire analysis is performed within the bootstrap framework.

---

> > > ### Author Response · Authors · 2025-11-22
> > > **Response to reviewer bN5G (part III)**
> > >
> > > **(4) Recommendations for LMArena.**
> > >
> > > We thank the reviewer for this suggestion. We have now included a discussion of our  recommendations for platforms like Chatbot Arena in light of our results (see section 6). Specifically, we recommend that platforms such as Chatbot Arena include robustness diagnostics as an additional measure of generalizability, alongside the confidence intervals they report. Sensitivity to worst-case data-dropping is often indicative of low signal-to-noise in the underlying data [2]; to help increase signal-to-noise, a few design-related improvements the arena could take include collecting richer forms of feedback beyond binary preferences (e.g., scalar or confidence-weighted ratings) and designing more discriminative prompts or incorporating a prompt-filtering system to identify and remove uninformative prompts. The original Chatbot Arena work performs topic-modeling of the prompts submitted to Chatbot Arena [1]. Their top-16 topics include "Poetry Writing Prompts" and "Movie Recommendations & Ratings." The subjective nature of such topics may make differentiation between top models less meaningful.
> > >
> > > **(5) Unbalanced Number of Matches.**
> > >
> > > We thank the reviewer for this observation. Although the highly unbalanced number of competitions may be one explanation for data-dropping sensitivity, we find that it need not be the case. For changing the top-ranked model on Chatbot Arena, we find that the models involved in the dropped subset have a sizable number of matches against one another, suggesting that data-dropping sensitivity cannot be attributed to a small sample size alone (see Figure 22 in Appendix J).
> > >
> > > [1] Chiang, W.-L., Zheng, L., Sheng, Y., Angelopoulos, A. N., Li, T., Li, D., Zhu, B., Zhang, H., Jordan, M., Gonzalez, J. E., et al. Chatbot arena: An open platform for evaluating llms by human preference. In Forty-first International Conference on Machine Learning, 2024.
> > >
> > > [2] Broderick, T., Giordano, R., and Meager, R. An automatic finite-sample robustness metric: When can dropping a little data make a big difference? arXiv preprint arXiv:2011.14999v1, 2020.

---

> > > > ### Comment · Reviewer_bN5G · 2025-11-27
> > > >
> > > > Thanks your detailed response and explanation! Good job!

---

### Official Review · Reviewer_BLi6 · 2025-11-10

**Soundness:** 3
**Presentation:** 2
**Contribution:** 2
**Rating:** 6
**Confidence:** 4

**Summary:**

This paper investigates how robust leaderboard rankings of LLMs are when based on human or AI preference data aggregated through the Bradley–Terry (BT) model. In particular, the authors consider an "intrinsic" notion of robustness (as opposed to, say, adversarial manipulation by a third party of the voting), where the minimal removal of some pairwise comparisons can alter the ranking of models---what the authors call top-k data dropping robustness in Definition 2.4. Building on the equivalence between data removal changing the top-k set and data removal changing the pairwise order between a top-k model and a model outside the top-k set (Proposition 3.1), the authors propose an approach to finding candidate data points whose removal can change the ranking. To that end, in the Appendix A, BT inference is formulated as logistic regression, and the authors use a first-order AMIP expansion to first identify the specific preferences most influential to ranking stability and then verify their effect through a recomputation of the ranking.

Applying this method to several open evaluation platforms, including Chatbot Arena, MT-Bench, Search Arena, Webdev Arena, and Vision Arena, the authors find that rankings can be extremely sensitive: on Chatbot Arena, removing just two out of nearly 60,000 human evaluations (around 0.003%) flips the top-ranked model. By contrast, MT-Bench shows much higher robustness, requiring the removal of several percent of data to change the top ranking. Human and LLM-as-a-judge evaluations exhibit comparable sensitivity levels.

**Strengths:**

- The general contextualization and introduction of the problem in Sections 1 and 2 are very clear and accessible. The authors make the paper easy to follow and introduce the necessary tools when needed (i.e., the main idea behind BT). The data-dropping setup and notation are also clearly communicated. Similarly, the theoretical results in the main text (most notably Prop. 3.1), while simple, are sound and well explained.

- The experiments conducted by the authors are comprehensive and effectively demonstrate that small percentages of dropped comparisons can have a significant impact on LLM rankings. This includes different evaluator models, different types of evaluators (human/LLM), and different dataset categories, providing strong support for the authors’ claims.

- The paper addresses a timely issue that can have important implications for LLM evaluation and practitioners in the field. The experimental results make the authors’ findings relevant.

**Weaknesses:**

Perhaps the key point I would like to inquire about is how the authors’ approach relates to other uncertainty quantification methods in the BT model. In particular, there are classical references that quantify uncertainty in BT coefficients, both in Bayesian and frequentist settings, for instance:

- Gao et al., "Uncertainty quantification in the Bradley-Terry-Luce model"
- Hunter "MM algorithms for generalized Bradley–Terry models"
- Leonard "An Alternative Bayesian Approach to the Bradley-Terry Model for Paired Comparisons"
- Negahban et al., "Rank Centrality: Ranking from Pair-wise Comparisons"

Such methods (including the estimated errors reported in platforms like LMArena, based on bootstrapping) are not discussed in the paper in detail and, a priori, also seem capable of quantifying the sensitivity of the ranking to data removal. Is this intuition correct? If so, I believe it would be helpful to explain how the authors’ methods relate or compare to such uncertainty quantification approaches.
In this direction, and since the authors conclude that the Chatbot Arena rankings are not robust, it would be interesting to interpret these results in light of the confidence intervals  reported by LMArena, which in most cases already suggest that the top-1 model is not estimated accurately.

Moreover:

- Interpretability and results. The authors claim that their method allows “inspection of these influential preferences.” While I agree that this is literally accurate, the authors’ method identifies a set of comparisons to drop, and one can only verify their influence after recomputing the BT ranking. In that sense, the interpretability benefits are limited.
In addition, I am having trouble understanding how the authors interpret the results regarding MT-Bench versus the rest of the LLM Arena data. The authors claim that the robustness of the MT-Bench ranking is partially due to the fact that “MT-Bench consists of 80 carefully designed multi-turn questions intended to differentiate models on core capabilities,” whereas LLM Arena prompts are less specific. Based on this reasoning, I would actually expect MT-Bench prompts to be more relevant for ranking models, and hence the ranking to be more sensitive to removals. However, the authors claim the opposite. A clarification here would be useful.

- Implementation. The authors do not provide a concrete algorithm in the main text summarizing their approach. I find this makes the implementation somewhat difficult to follow and would benefit from further clarification. In line 280, the authors provide a textual description of the process; however, this description seems to omit key information, such as how the method selects the reweighting vectors. Moving such details from the appendix to the main text could therefore be beneficial.
As a consequence of omitting this information, some of the authors’ claims are confusing. For instance, in line 188: “Notice that both Equation (5) and Equation (7) are nontrivial to directly verify; to check directly, we have to test out dropping all possible small-fraction subsets of the arena, a combinatorial operation that is computationally intractable in practice.” The authors could explain more explicitly how their method addresses this issue.

- Performance of the AMIP.  To better understand the efficiency of the authors’ method---since it requires recomputing the BT inference for the chosen vector w---it would be beneficial to show how this search method for w performs. For instance, the authors could report, in the main text, the fraction of vectors w that correctly lead to a different ranking.

**Questions:**

I believe that while most of the technical machinery and setup of the paper are clear, the overall presentation could be made more concise. In particular, some presentation choices feel redundant:
- On line 269, the authors write, “For a candidate pair of players, (i, j), recall that we assumed without loss of generality…”. This condition was already introduced in line 247. Twenty-two lines seem close enough for this repetition to be unnecessary.
- Equations (8) and (5) are exactly the same. While I understand that the context differs---since Eq. (5) refers to the two-player case---this raises the question of whether that case should have been treated separately at all.


Some other minor comments/suggestions:
- As far as I understand, Figure 2 is not an original result of the paper but rather summarizes open-access information. However, it currently occupies about one-third of a page in the main text. Perhaps replacing it with a smaller table or moving it to the appendix would be more appropriate.
- In line 193/194, I believe "Theorem 2.4" should read "Definition 2.4"
- In line 211, the sentence “Then, it must be that S is the top-k set, or S = K(w)” reads as if there were a dichotomy, whereas I believe the authors simply want to clarify a single condition---that S is the top-k set. Avoiding “or” would make this clearer.
- In line 233, the authors use the term “teams” for the first time, but it is unclear why this terminology is introduced, given that it is not used again later.

---

> ### Author Response · Authors · 2025-11-22
> **Response to reviewer BLi6 (part I)**
>
> We thank the reviewer for highlighting the clarity of our setup and the comprehensiveness of our experiments. Below, we provide detailed responses to each of the reviewer’s questions and comments. We refer the reviewer to the general reply for a high-level overview of the new additions.
>
> **(1) Relation to classical BT uncertainty quantification and results in light of the bootstrap-based rankings reported on LMArena.**
>
> We thank the reviewer for inquiring about the difference between frequentist [3, 4] and Bayesian [5] Bradley–Terry (BT) uncertainty quantification methods and sensitivity of the estimated rankings to the dropping of small, worst-case subsets of data. We have added the reviewer’s suggested citations into Appendix A.2 of our revised manuscript, including an extended discussion on the distinction between worst-case data-dropping sensitivity and classical uncertainty quantification methods. While frequentist uncertainty quantification methods, such as the bootstrap, ask how a reported statistic might vary if the sample were re-generated, capturing randomness in the data-generating process, the AMIP targets sensitivity on a single, fixed dataset.  This focus on a single sample differs in spirit from the variability across “counterfactual worlds” that frequentist uncertainty quantification methods measure. In this sense, the two approaches answer complementary questions about the stability of sample-based conclusions: the confidence interval measures sampling uncertainty, while a worst-case data-dropping check examines whether the ranking is driven by a very small fraction of the points in the sample. In fact, previous works such as [5] have pointed to instances where data analyses are both statistically significant and AMIP-sensitive.
>
> To directly address the reviewer’s concerns, we added an experiment where we applied our sensitivity analysis to the bootstrap-based rankings reported by LMArena [1]. Specifically, we computed the original bootstrap-based rankings (defined both in Chiang et al. and now in our paper’s Appendix A.1),  removed the influential comparisons identified by our method, and then recomputed the bootstrap-based rankings. The results of this new set of experiments are summarized in Table 2 in Appendix A.1. Even under the bootstrap-based rankings, we continue to observe that many arenas are sensitive to worst-case small-fraction data-dropping. The reviewer points out that the existing bootstrap-based rankings “already suggest that the top-1 model is not estimated accurately.” With this new set of experiments, we surface arenas where the bootstrap-based ranking does output a single top-ranked model, but upon small-fraction data dropping, the model becomes no longer the sole top-ranked model (see Figure 7 in Appendix A.1). This shows that the non-robustness identified by AMIP is not an artifact of ignoring statistical uncertainty already captured by confidence intervals; the ranking flip persists even when the entire analysis is performed within the bootstrap framework.
>
> Although Bayesian uncertainty quantification methods also operate under the case of a single, fixed dataset, past work has demonstrated that data analyses can be both statistically significant in the Bayesian sense (credible interval does not include zero) and still sensitive to worst-case data dropping (see Bayesian hierarchical model case study in Section 4.4 of [2]).  So, analogous to the frequentist case, the AMIP again represents a different and complementary check.
>
> [1] Chiang, W.-L., Zheng, L., Sheng, Y., Angelopoulos, A. N., Li, T., Li, D., Zhu, B., Zhang, H., Jordan, M., Gonzalez, J. E., et al. Chatbot arena: An open platform for evaluating llms by human preference. In the Forty-first International Conference on Machine Learning, 2024.
>
> [2] Broderick, T., Giordano, R., and Meager, R. An automatic finite-sample robustness metric: When can dropping a little data make a big difference? arXiv preprint arXiv:2011.14999v1, 2020.
>
> [3] Gao, Chao, Yandi Shen, and Anderson Y. Zhang. "Uncertainty quantification in the Bradley–Terry–Luce model." Information and Inference: A Journal of the IMA 12.2 (2023): 1073-1140.
>
> [4] Hunter, David R. "MM algorithms for generalized Bradley-Terry models." The annals of statistics 32.1 (2004): 384-406.
>
> [5] Leonard, Tom. "An alternative Bayesian approach to the Bradley-Terry model for paired comparisons." Biometrics (1977): 121-132.

---

> > ### Author Response · Authors · 2025-11-22
> > **Response to reviewer BLi6 (part II)**
> >
> > **(2) Interpretability of results and explanation for the robustness of MT-Bench.**
> >
> > We thank the reviewer for this question. While MT-Bench’s prompts are indeed higher quality and may thus be thought of as more relevant for resulting rankings, the key for robust rankings is not the absolute importance of each prompt, but the signal-to-noise ratio with which the prompts collectively separate the BT scores.
> >
> > We suspect that crowdsourced arenas like Chatbot Arena exhibit a lower signal-to-noise ratio compared to MT-bench, as crowdsourced prompts may be less informative in discriminating model capabilities and thus unable to strongly distinguish top models. Annotators on crowdsourced platforms may also be more likely to provide noisy, or mislabeled, preferences. The original AMIP paper provides a theoretical explanation (See section 3 of [1]) suggesting that sensitivity to worst-case data dropping may stem from low signal-to-noise ratio in the inference problem. Low signal-to-noise ratio in the annotated preferences in turn makes the rankings on Chatbot Arena fragile: dropping even a few arbitrary wins or losses can push one model above another. In contrast, MT-Bench consists of carefully designed prompts explicitly aimed at differentiating models on core skills (e.g., math, reasoning, and writing). These higher-quality prompts may provide a more consistent discriminative signal, which leads to robust rankings. This is indeed what we observe in our results. The BT-score margins on MT-Bench (Figure 8) are larger than those on Chatbot Arena (Figure 3), which may contribute to why MT-Bench requires removing a larger fraction of preferences before its top rankings change. Finally, all robustness numbers in the paper are reported as fractions of the total data rather than raw counts, so the comparison between MT-Bench and large crowdsourced platforms is normalized for dataset scale.
> >
> > **(3) Implementation.**
> >
> > We thank the reviewer for this suggestion. We have now provided a concrete algorithm at the end of Section 3; the algorithm includes a clear description of how our method selects the reweighting vector via a sorting of influence scores.
> >
> > **(4) Performance of the AMIP.**
> >
> > We thank the reviewer for this suggestion. We would first like to clarify that the computational bottleneck in worst-case data-dropping comes in the combinatorial search over data subsets, not in recomputing the BT scores for a chosen vector, w. And our method sidesteps the combinatorial search by using AMIP to produce a candidate weight vector w (now described in the blue text in Section 3), after which the method performs a single BT refit on the candidate vector w (e.g., without the dropped subset) to verify whether a ranking flip actually occurs. This BT model (i.e. a logistic regression model) takes less than a second to fit even for the largest arena we examine (0.80 seconds for Chatbot Arena run on the full dataset).
> >
> > We also endeavored to directly address the reviewer’s question about the effectiveness of AMIP at identifying a subset to drop that indeed leads to changes in ranking. In particular, for each of our nine data analyses, we can use the machinery of our method to return a w corresponding to a smallest data subset that can be dropped to change the top-1 ranking. The machinery of our method also returns an estimate (before re-running the BT model) for whether the top-1 ranking is changed. In the revision of our manuscript (Table 4 in Appendix G), we report (The number of cases where the estimate with this w vector accurately reflects a change in the top ranking​) / (Total number of arenas tested for top-1 robustness). We find that all identified w-vectors lead to a true change in ranking. And we find that this result holds even when the dropped subset is greater than $\lfloor \alpha N \rfloor$ of the data (even though the original AMIP [1] makes no claims to an accurate identification of a decision-changing w in this case). However, this does not mean that there are no cases where AMIP fails to surface a vector w that leads to a change in ranking (i.e., false negatives are possible).
> >
> > [1] Broderick, T., Giordano, R., and Meager, R. An automatic finite-sample robustness metric: When can dropping a little data make a big difference? arXiv preprint arXiv:2011.14999v1, 2020.

---

> > > ### Author Response · Authors · 2025-11-22
> > > **Response to reviewer BLi6 (part III)**
> > >
> > > **Questions, minor comments, and suggestions:**
> > >
> > > *On line 269, the authors write, “For a candidate pair of players, (i, j), recall that we assumed without loss of generality…”. This condition was already introduced in line 247. Twenty-two lines seem close enough for this repetition to be unnecessary.*
> > >
> > > **A**: We thank the reviewer for this suggestion. We have now left out the second mention.
> > >
> > > *Equations (8) and (5) are exactly the same.*
> > >
> > > **A**: We thank the reviewer for this great point. We have since removed Eq. (5) from the manuscript as well as the discussion of Two-player Arenas. We now jump more directly into the general definition for an M-player arena.
> > >
> > > *As far as I understand, Figure 2 is not an original result of the paper but rather summarizes open-access information.*
> > >
> > > **A**: We thank the reviewer for this suggestion. We now display the original Chatbot Arena rankings in Figures 3 and 4 of Appendix A.1.
> > >
> > > *In line 193/194, I believe "Theorem 2.4" should read "Definition 2.4"*
> > >
> > > **A**: We thank the reviewer for catching this typo. We have now fixed this in the manuscript accordingly.
> > >
> > > *In line 211, the sentence “Then, it must be that S is the top-k set, or S = K(w)” reads as if there were a dichotomy, whereas I believe the authors simply want to clarify a single condition---that S is the top-k set. Avoiding “or” would make this clearer.*
> > >
> > > **A**: For clarity, we have replaced “or” with “i.e.”.
> > >
> > > *In line 233, the authors use the term “teams” for the first time, but it is unclear why this terminology is introduced, given that it is not used again later.*
> > >
> > > **A**: We thank the reviewer for catching this. We have now changed “teams” to “players” for consistency.

---

> ### Comment · Reviewer_BLi6 · 2025-11-22
>
> I would like to thank the authors for their detailed and complete response. Below, I provide some additional feedback on two points:
>
> - **Relation to classical BT uncertainty quantification and the bootstrap-based rankings reported on LMArena.**
> I find the additions in Appendix A very clear. The discussion is helpful for readers in understanding the connections and distinctions with bootstrap-based ranking methods. I also find the new results in Table 2 quite interesting.
>
> - **Performance of the AMIP.**
> I find the new evidence presented in Table 4 in Appendix G convincing and impressive in demonstrating the efficacy of the proposed method. For readers, it substantially clarifies the experimental results of the paper. I might have otherwise suggested comparing the authors’ method with a uniform baseline that randomly drops comparisons, but the authors have already included such an experiment in Appendix Table 3. I believe this new evidence was necessary and significantly strengthens the contribution of the method.
>
> I believe the new modifications of the paper significantly improve it, and will update my score accordingly.

---

### Official Review · Reviewer_yXBg · 2025-11-11

**Soundness:** 3
**Presentation:** 3
**Contribution:** 3
**Rating:** 8
**Confidence:** 4

**Summary:**

The paper focuses on the sensitivity of LLM ranking platforms based on crowdsourced pairwise comparisons of their responses. Specifically, the authors consider a ranking setting based on the Bradley-Terry model, and they develop a method to identify a small number of pairwise comparisons whose absence would alter the top-k set in a ranking. They then use this method to analyze the sensitivity of the resulting rankings using comparison data from multiple popular crowdsourced platforms such as Chatbot Arena and MT-bench. The results in the paper indicate that removing a very small number of pairwise comparisons is sufficient to alter the resulting rankings.

**Strengths:**

The main strengths of the paper are as follows:
1. It studies a timely problem, as benchmarking platforms such as LM Arena become the industry standard for evaluating the performance of large language models. Hence, the results will be of interest to the ICLR community.
1. It develops a relatively simple method building upon prior work in statistics for identifying small sets of pairwise comparisons that significantly affect the resulting model rankings.
1. Its analysis reveals a surprising insight that a very small number of pairwise comparisons (far less than 1%) is sufficient to lead to different conclusions regarding which model is the top performer according to thousands of pairwise comparisons by human users (Table 1)

**Weaknesses:**

I believe that the paper has some room for improvement in terms of its presentation of certain definitions and results and its experimental evaluation. Specifically:
* The definitions in page 4 are somewhat confusing. It is unclear why there is a need to introduce Definition 2.2 regarding top-1 robustness in two-player arenas, as it is immediately followed by the more general definition of top-k robustness in arenas with more than two players. Definition 2.2 doesn't seem to be used anywhere. Moreover, this definition seems identical to what the authors call "pairwise robustness" of scores in page 5, although there it is not presented explicitly as a definition. All these redundant definitions just make it harder for the reader to focus on the core flow of the presentation.
* The method that the authors introduce evaluates the robustness of a top-k set in a ranking by separately evaluating the robustness of pairwise model orderings according to their estimated scores. This is explained in lines 238-242, but it is not explicitly proved that the two robustness checks are equivalent. Instead, the authors provide a proof for Proposition 3.1, which seems rather obvious (i.e., that a set of size k whose values are greater than all the values not in that set is the top-k set) and doesn't seem like a proof worth including in the main body of the paper. I believe it would be helpful for the presentation if the authors could restructure and clarify that part of the paper a bit.
* My understanding is that the method introduced by the authors is valid when it finds a set of pairwise comparisons that would alter a ranking (i.e., the ranking would indeed be different), but I think it can still suffer by false negatives due to the fact that the method solves a combinatorial problem approximately via continuous optimization. In other words, if the method fails to find a set of pairwise comparisons that alters the ranking, this does not necessarily mean that no such set exists and the ranking is robust. Could the authors comment on that? For example, can we say with certainty that the 3 datasets in gray in Table 1 are truly robust? It would have also been interesting to see some experiments showcasing failure cases of the method itself, as the current experimental evaluation focuses solely on the sensitivity of rankings and does not present any analysis of the method itself.
* I think the paper would have been richer if the authors performed a more thorough qualitative analysis of the pairwise comparisons that their method identifies as critical for a ranking. For example, I found the fact that removing only 2 data points was sufficient to change the winner in the Chatbot Arena ranking very intriguing. However, other than the fact that these 2 data points corresponded to battles where the top performing model lost by a model placed far lower in the ranking, there aren't any further insights provided in the paper that could inform how to design more robust ranking methods in the future. For example, was there any pattern in the rating behavior of the users who submitted those critical comparisons? Is the sensitivity of rankings always affected largely by such "abnormal" pairwise comparisons? Is there any systematic way to filter them out?

**Questions:**

I don't have additional questions other than what I already brought up under "Weaknesses". It would be great if the authors could elaborate on those points in their rebuttal.

---

> ### Author Response · Authors · 2025-11-22
> **Response to reviewer yXBg (part I)**
>
> We thank the reviewer for the thorough review and valuable feedback and for highlighting the timeliness of this work and the simplicity of our method. Below, we provide detailed responses to each of the reviewer’s comments. We refer the reviewer to the general reply for a high-level overview of the new additions.
>
> **(1) Definitions on Page 4.**
>
> We appreciate the reviewer’s comment to improve the flow of the paper. We have removed Definition 2.2 and the discussion of Two-player Arena. We now jump directly into the general definition for an M-player arena.
>
> **(2) Equivalence Between Checking Robustness of Pairs of Players and Checking Robustness of the Top-k Set.**
>
> We thank the reviewer for this feedback and have clarified the equivalence between these two robustness checks in the paper (see blue text at the beginning of Section 3). In short, the results given by Proposition 3.1 tell us that we can check top-$k$ robustness by checking pairwise robustness of all models inside the top-$k$ set against all models outside of this set. Because, in the case that there exists a pair of models (one inside and one outside) whose rankings flip, then the top-$k$ set has changed, which means the arena is non-robust. In the case that there does not exist at least one such pair of models whose rankings can be flipped upon dropping a small fraction of preferences, then the top-$k$ set remains the same, i.e., the arena is top-$k$ robust. The original writing did not make the connection between Proposition 3.1 and the equivalence of the two checks, which we have now done following the reviewer’s suggestion. We have also moved Proposition 3.1 into the appendix (see Appendix B).
>
> **(3) Possibility of False Negatives.**
>
> We thank the reviewer for raising this point. We fully agree that any approximation to an underlying combinatorial robustness problem must be understood through both its successes and its potential failures and so have added a discussion on the possibility of false negatives in the appendix (see the new Appendix H for an extended discussion). We also added a statement about false negatives as well as a reference to the discussion in Appendix H into the methods section of the manuscript (see blue text in Section 3, directly above the paragraph about runtime).
>
> **(4) Qualitative Analysis.**
>
> We thank the reviewer for this feedback. In Appendix D, we have added a qualitative analysis on the surfaced response pairs, in which we pass the responses to a strong judge model, GPT-5.1, with a system prompt requesting a summary of the two responses, their similarities and differences, and a difficulty rating indicating how distinct the responses are (see qualitative comparison in Appendix D). In both surfaced examples, GPT-5.1 judges the pair of responses to be easy to differentiate and consistently prefers the opposite response from the original rater (e.g., "A typical user would likely find Response A significantly more helpful, complete, and actionable than Response B," and "A typical user would quickly notice that only Response A directly addresses the C++-project requirement."). This makes sense, as both matches are those in which the much lower-scoring model is preferred to the top-ranked model. Thus, one might interpret the influential subsets we identify as "outlier" judgments, cases where the annotator’s preference deviates from what a "typical" user might select.
>
> We believe, however, that diagnosing the root cause of non-robustness lies beyond the scope of this current work. We note that the BT model’s estimation procedure does not use any information about prompt content; it depends only on the BT scores of the two models involved in each match, which means it is possible for dropped subsets to be non-unique. We have added this additional note in Appendix D (blue text). For this reason, we find it more meaningful to describe identified influential comparisons through the lens of the score differences among models involved in the dropped comparisons. Across all 36 matches surfaced while checking top-k robustness on Chatbot Arena, every dropped preference was a clear win or loss (no ties), and every outcome aligned with the direction required to flip the ranking. We have added this discussion into Appendix C (see blue text).

---

### Author Response · Authors · 2025-11-22
**Rebuttal summary and changes in the new version of the paper**

We thank the reviewers for all their valuable feedback and constructive suggestions. We are encouraged that reviewers found the research question to be novel, the method simple, the findings surprising, and the work a critical piece of scientific auditing for the LLM community. The reviewers’ constructive feedback has helped us strengthen and add clarity to the paper. Across the rebuttal, we made additions responding to all major reviewer concerns.

*The key updates are outlined below:*

**1. Uncertainty Quantification Section.**

a. We added a new set of experiments running our robustness check on the bootstrap confidence-interval-based rankings. We report on these new findings in both **Section 4.2** and **Appendix A.1**.

b. We added a section clarifying the relationship between worst-case data-dropping sensitivity and classical uncertainty quantification (see **Appendix A.2**).

c. We added a uniform-at-random data-dropping experiment to contrast with our experiments on worst-case data dropping (see **Appendix A.3**).

**2. AMIP Performance.**

a. To address a reviewer question on the effectiveness of AMIP at identifying a subset to drop that indeed leads to a change in ranking, we added an experiment to verify that all AMIP-identified subsets led to ranking flips (see **Appendix G**).

b. We added an extended discussion on the possibility of false negatives (see **Section 3** and **Appendix H**).

**3. Qualitative Analysis of Surfaced Preferences.**

a. We added a qualitative analysis of the two surfaced Chatbot Arena responses (see **Section 4.4** and **Appendix F**) and discussed why a generalizable qualitative analysis may be beyond the scope of our work (see responses).

**4. Structural Improvements to the Setup and Methods Sections.**

a. We added a concrete algorithm outlining the full robustness check to the main text (see **Section 3**).

b. We improved the flow of the setup and methods sections (**Sections 2** and **3**) by doing away with redundant definitions and moving details into the appendix.

---

### Meta-Review · Area_Chair_XFaq · 2026-01-13

**Summary:**

This paper asks a clean and important auditing question for BT-style LLM leaderboards: can the top-k ranking flip after removing a worst-case tiny fraction of preference data? It proposes a fast, easy-to-run robustness check (AMIP-style influence approximation + exact re-fit verification) and applies it to major arenas (Chatbot Arena and derivatives), finding surprisingly high sensitivity at the very top (e.g., flipping the top-1 with only a handful of preferences removed).


Overall, I like this paper and the contribution is strongest as a diagnostic / auditing tool plus a set of empirical findings that the community will appreciate.

**Reviewer Concerns:**

The main reviewer concerns were about execution details and framing rather than the core idea:

1. some early presentation issues (redundant definitions, unclear link between pairwise flips and top-k robustness),
2. how to interpret worst-case dropping vs classical uncertainty quantification (bootstrap / CIs),
3 the possibility of false negatives and a desire for better qualitative/contextual understanding of the “critical” preferences.

I think the authors provided a strong rebuttal was strong to address most of the these concerns.

Remaining limitations are reasonable: the paper is a diagnostic (not a “fix”), and deeper causal explanation of why specific preferences are influential is largely left for follow-up work.

**Reviewer Scores:**

I think all reviewers would likely keep or bump up their score if a rebuttal had completed.

---

### Decision · Program_Chairs · 2026-01-26

Accept (Poster)